



# Bulk hydrometeor optical properties for microwave and sub-mm radiative transfer in RTTOV-SCATT v13.0

Alan J. Geer[1], Peter Bauer[1], Katrin Lonitz[1], Vasileios Barlakas[2], Patrick Eriksson[2], Jana Mendrok[2,3], Amy Doherty[4], James Hocking[4], and Philippe Chambon[5]

[1]ECMWF, Shinfield Park, Reading, RG2 9AX, UK
[2]Department of Space, Earth and Environment, Chalmers University of Technology, Gothenburg, Sweden
[3]Now at Deutscher Wetterdienst, Offenbach, Germany
[4]Met Office, UK
[5]CNRM, Université de Toulouse, Météo-France, CNRS, Toulouse, France

**Correspondence:** Alan Geer (alan.geer@ecmwf.int)

**Abstract.** Satellite observations of radiation in the microwave and sub-mm spectral regions (broadly from 1 to 1000 GHz) can have strong sensitivity to cloud and precipitation particles in the atmosphere. These particles (known as hydrometeors) scatter, absorb and emit radiation according to their mass, composition, shape, internal structure, and orientation. Hence, microwave and sub-mm observations have applications including weather forecasting, geophysical retrievals and model validation. To

simulate these observations requires a scattering-capable radiative transfer model and an estimate of the bulk optical properties of the hydrometeors. This article describes the module used to integrate single-particle optical properties over a particle size distribution (PSD) to provide bulk optical properties for the Radiative Transfer for TOVS microwave and sub-mm scattering code, RTTOV-SCATT, a widely-used fast model. Bulk optical properties can be derived from a range of particle models including Mie spheres (liquid and frozen) and non-spherical ice habits from the Liu and Atmospheric Radiative Transfer

Simulator (ARTS) databases, which include pristine crystals, aggregates and hail. The effects of different PSD and particle options on simulated brightness temperatures are explored, based on an analytical two-stream solution for a homogeneous cloud slab. The hydrometeor scattering "spectrum" below 1000 GHz is described, along with its sensitivities to particle composition (liquid or ice), size and shape. The optical behaviour of frozen particles changes in the frequencies above 200 GHz, moving towards an optically thick and emission-dominated regime more familiar from the infrared. This region is previously little

explored but will soon be covered by the Ice Cloud Imager (ICI).



# 1 Introduction

Observations of electromagnetic radiation in the microwave and sub-millimetre (sub-mm) spectral regions [1] can have strong sensitivity to hydrometeors, i.e. cloud and precipitation particles in the atmosphere. The primary sensitivity is to the mass
and composition of the particles but there is also information on a range of microphysical characteristics. These observations are used for improving our understanding of cloud physics, for cloud and precipitation retrievals (Skofronick-Jackson et al., 2018), for model evaluation (Ori et al., 2020), and for all-sky data assimilation in operational weather forecasting (Geer et al., 2017, 2018). Extracting the physical information from these observations is an inverse problem (e.g. Rodgers, 2000) at the core of which is the forward model, that maps from the physical state to the observables, in our case microwave/sub-mm
radiances or radar reflectivity. Hydrometeors absorb, emit and scatter microwave and sub-mm radiation according to their mass, composition (potentially including water, ice and air), shape, internal structure, and orientation (e.g. Eriksson et al., 2015; Schrom and Kumjian, 2018; Ekelund et al., 2020a).

The optical properties of a single particle can be computed with approaches such as the Mie theory or, for non-spherical particles, the discrete dipole approximation (DDA, Draine and Flatau, 1994). To represent the optical properties of a layer
of cloud, the properties of every particle in that cloud need to be considered. This is usually done by assuming knowledge of the particle size distribution (PSD), habits and orientations, and integrating across the size spectrum of the hydrometeors. This produces the "bulk hydrometeor optical properties" that are the necessary input to a model for radiative transfer or radar propagation that is used to forward model the observed quantity, i.e. radiance or radar reflectivity. This work describes, from a scientific point of view, a widely-used software tool for generating lookup tables of bulk hydrometeor optical properties
for use in the forward modelling, and hence for use in numerous applications in atmospheric physics, retrievals, and weather forecasting.

The tool to be described is known as the hydrometeor optical table (hydrotable) generator and is a self-standing component of the Radiative Transfer for TOVS (RTTOV, Saunders et al., 1999, 2018) fast model. RTTOV provides tools to simulate observations from over 80 spaceborne sensors operating from the microwave to the visible parts of the spectrum, and it has
over 1000 registered users including operational centres and scientists worldwide. This article refers to RTTOV version 13.0, released in November 2020 (Saunders et al., 2020). The hydrotable generator supports the microwave and sub-mm component of RTTOV, known as RTTOV-SCATT (Bauer et al., 2006), which provides fast modelling for dozens of spaceborne radiometers and radars. For example this includes the microwave imager and radar onboard the Global Precipitation Mission (GPM, Skofronick-Jackson et al., 2018) and numerous research and operational sensors operated by space agencies worldwide. A
focus of current development is the future Ice Cloud Imager (ICI, Buehler et al., 2007; Eriksson et al., 2020, launch planned for 2024), which will be the first operational mission to provide measurements above 200 GHz and into the sub-mm. This spectral range is expected to be more sensitive to cloud ice than frequencies currently used.

The original code for the hydrotable generator came out of the work of Bauer (2001) and was brought into RTTOV with the addition of RTTOV-SCATT (Bauer et al., 2006). The hydrotable generator was initially extended to simulate ice cloud

---

[1] Microwave is 300 MHz to 300 GHz; sub-mm is above 300 GHz and below the infrared





signatures at higher frequencies (e.g. 183 GHz) by Doherty et al. (2007) and then much revised by Geer and Baordo (2014) to
      incorporate the database of non-spherical frozen particles from Liu (2008). The move to representing snow as a non-spherical
      particle (rather than a Mie soft sphere) unlocked the use of the higher microwave frequencies in weather forecasting (Geer
      and Baordo, 2014; Geer et al., 2017). The code has been much updated for version 13.0 of RTTOV, with improved models
      of water permittivity (Lonitz and Geer, 2019), a first treatment of hydrometeor orientation (Barlakas et al., 2020), and the
addition of the Atmospheric Radiative Transfer Simulator (ARTS) scattering database with a wider range of frozen particles,
      such as aggregates and hail (Eriksson et al., 2018). There has also been a major expansion of the available PSDs (adding
      e.g. McFarquhar and Heymsfield, 1997; Petty and Huang, 2011; Heymsfield et al., 2013). And although the tool is fully
      configurable, the default configuration is widely used, so the microphysical choices in that default configuration have been
      carefully selected. Since the physical global properties of hydrometeors are not well known, the microphysical settings for
v13.0 were updated by parameter estimation, based on the fit between real observations and those simulated from a weather
      forecasting model (Geer, 2021b). In that work, a particular effort was made to better represent ice hydrometeors in preparation
      for ICI: these are now represented by a large plate aggregate for snow, a column for graupel, and a large column aggregate
      for ice cloud. The ice cloud PSD has also been updated, noting that commonly-used PSDs appear to generate too many large
      particles to properly represent the "ice cloud" category in global models. For v13.0, the code has been moved to SI units (with
a few exceptions) and away from the mix of centimetres-grams-seconds (CGS) and other unit systems employed in the past by
      the microwave community. The core integration over particles has also been revised, uncovering a number of detailed issues
      on the way. On the technical side, the code is primarily Fortran. It is able to generate lookup tables for over 100 channels and
      34 instruments in a few minutes on a multi-core workstation.

      The process of integrating single-particle optical properties over a PSD is a standard task in any radiative transfer package
with cloud and precipitation capabilities, such as ARTS (Buehler et al., 2018), the Community Radiative Transfer Model
      (CRTM, https://github.com/JCSDA/crtm), the Passive and Active Microwave radiative TRAnsfer tool (PAMTRA, Mech et al.,
      2020) and the radar and lidar forward simulator ZmVar (Di Michele et al., 2012; Fielding and Janiskova, 2020), which has also
      evolved from the original code of Bauer (2001). However the hydrotable generator in RTTOV is one of the most comprehensive
      available, and certainly it is widely used. This work aims to be both a scientific user guide to the hydrotable generator in RTTOV
and also a helpful reference for users of similar tools. The detailed technical user guide is included as a readme file along with
      the software, which is the ultimate reference. Here we concentrate on the broader science and on providing guidance on the
      physical choices available. Section 2 overviews the tool, the sources of single-particle optical properties, and finally the bulk
      optical properties produced by the tool. Section 3 describes the methods in more detail, focusing on recent developments,
      such as the PSDs, that have not been covered elsewhere. Section 4 introduces a standardised framework for comparing bulk
optical properties, based on an analytic solution of the two-stream equations for a homogeneous cloud. This helps illustrate and
      compare the available physical options and to overview the basic properties of hydrometeors in the microwave and sub-mm
      regions. The conclusion looks to future developments.





## 2 Overview of hydrotable generator

Bulk optical properties are the integrated contributions of the optical properties of all individual cloud or precipitation particles
within a unit volume. It is assumed that the average single-particle optical properties are known as a function of the particle
size, here the geometric diameter $D_{\mathrm{g}}$, which is the maximum dimension in the case of non-spherical particles. The number of
particles of each size is described by the particle size distribution (PSD) $n'_{\mathrm{g}}(D_{\mathrm{g}})$ which gives the number density of particles
per unit of particle diameter [$\mathrm{m}^{-3}\,\mathrm{m}^{-1} = \mathrm{m}^{-4}$]. The bulk optical properties can then be computed by numerically integrating
the single-particle properties over the PSD. For example the bulk extinction coefficient $\beta_{\mathrm{e}}$ [$\mathrm{m}^2\,\mathrm{m}^{-3} = \mathrm{m}^{-1}$] can be computed
by integrating the single-particle extinction cross-section $\sigma_{\mathrm{e}}(D_{\mathrm{g}})$ [$\mathrm{m}^2$] as follows:

$$\beta_{\mathrm{e}} = \int_{D_{\min}}^{D_{\max}} \sigma_{\mathrm{e}}(D_{\mathrm{g}}) n'_{\mathrm{g}}(D_{\mathrm{g}})\, dD_{\mathrm{g}}. \tag{1}$$

The integration is done over a size range $D_{\min}$ to $D_{\max}$ that will be discussed in Sect. 3.2. Note that in this work the prime on
the PSD notation $n'_{\mathrm{g}}(D_{\mathrm{g}})$ indicates that it has been rescaled to account for numerical integration issues and the limited size
range, a process referred to as renormalisation (Sec. 3.2.2, Eq. 17).

To represent scattering in active and passive microwave/sub-mm radiative transfer in a fast model like RTTOV-SCATT
requires also the bulk scattering and backscatter coefficients $\beta_{\mathrm{s}}$ and $\beta_{\mathrm{b}}$, both in [$\mathrm{m}^{-1}$], and the dimensionless bulk asymmetry
parameter $g$ which summarises the mean direction of scattering (strictly, $g$ is the phase-function weighted mean of the cosine
of the scattering angle; see Petty, 2006, for full definitions). These bulk properties are computed from the single-particle
scattering and backscatter cross-section $\sigma_{\mathrm{s}}(D_{\mathrm{g}})$ and $\sigma_{\mathrm{b}}(D_{\mathrm{g}})$ [$\mathrm{m}^2$] and single-particle asymmetry $g_{\mathrm{single}}(D_{\mathrm{g}})$ [dimensionless],
again by integrating over the PSD:

$$\beta_{\mathrm{s}} = \int_{D_{\min}}^{D_{\max}} \sigma_{\mathrm{s}}(D_{\mathrm{g}}) n'_{\mathrm{g}}(D_{\mathrm{g}})\, dD_{\mathrm{g}}; \tag{2}$$

$$\beta_{\mathrm{b}} = \int_{D_{\min}}^{D_{\max}} \sigma_{\mathrm{b}}(D_{\mathrm{g}}) n'_{\mathrm{g}}(D_{\mathrm{g}})\, dD_{\mathrm{g}}; \tag{3}$$

$$g = \frac{1}{\beta_{\mathrm{s}}} \int_{D_{\min}}^{D_{\max}} g_{\mathrm{single}}(D_{\mathrm{g}}) \sigma_{\mathrm{s}}(D_{\mathrm{g}}) n'_{\mathrm{g}}(D_{\mathrm{g}})\, dD_{\mathrm{g}}. \tag{4}$$

Note the bulk asymmetry is a weighted average using the scattering cross section.

By evaluating these integrals many times, lookup tables are generated as a function of temperature, water content and
channel; if required a simple integration across the spectral response function of the instrument is also performed. The lookup
tables are written out as data files ("hydrotables"), one for each target instrument, containing the following representation of
the bulk optical properties:





**Table 1.** Default 5-hydrometeor settings

| Hydrometeor placeholder | Scattering type | Particle shape | PSD | MGD parameters | | | | Extended size range | Integration type | Max. renorm. |
| --- | --- | --- | --- | --- | --- | --- | --- | --- | --- | --- |
| | | | | $N_0$ | $\mu$ | $\Lambda$ | $\gamma$ | | | |
| Rain | Mie | sphere | MGD | $8\times10^6$ | 0 | free | 1 | F | New | 0.05 |
| Snow | ARTS | large plate aggregate | F07 T | - | - | - | - | F | Old | 0.5 |
| Graupel | ARTS | column | F07 T | - | - | - | - | F | Old | 0.5 |
| Cloud water | Mie | sphere | MGD | free | 2 | $2.13\times10^5$ | 1 | F | New | 0.001 |
| Cloud ice | ARTS | large column aggregate | MGD | free | 0 | $1\times10^4$ | 1 | T | New | 0.001 |

Acronyms for the particle size distributions (PSDs) and parameters and units of the Modified Gamma Distribution (MGD) are defined in Sec. 3.1.

- the bulk extinction coefficient $\beta_e$. In an exception to the SI policy used elsewhere, the units of the extinction coefficient are [km$^{-1}$];

- the dimensionless bulk single scattering albedo (SSA) $\omega_0 = \beta_s/\beta_e$;

- the dimensionless bulk asymmetry parameter $g$;

- if the targeted sensor is a radar, also the bulk radar reflectivity factor $Z = (10^{18}/z_0)\beta_b$, in [mm$^6$ m$^{-3}$]. See Appendix A for full definition.

The data files contain bulk optical properties for a set of possible hydrometeor types. The default configuration of the table generator is given in Table 1, which provides 5 hydrometeor types representing rain, "snow" (referring to precipitating particles in stratiform cloud), "graupel" (referring to all ice particles in convective cores), cloud water and cloud ice (referring to 120 suspended frozen particles). This maps onto typical hydrometeor representations in global forecast models. However the total number of hydrometeor types in RTTOV-SCATT is unlimited and this could be used to build up more complex representations (for example, there could be different hydrometeor types for tropical and extratropical ice cloud). Each hydrometeor type is defined by a set of physical options, with the main options illustrated in Table 1; these will be described in more detail in the rest of this article. The default settings for frozen hydrometeors were obtained from a multi-dimensional parameter search 125 in order to produce the best fits between ECMWF modelled brightness temperatures and Special Sensor Microwave Imager Sounder (SSMIS) observations (Geer, 2021b). The settings for rain and cloud water have been inherited from Bauer (2001).

Each hydrometeor type needs to be associated with one of the "placeholder" types listed in Table 2. This gives the hydrometeor a descriptive name and indicates whether it is frozen or liquid. It also associates a density and size range $D_{min}$ to $D_{max}$, which are mainly relevant when the Mie sphere approximation is used to compute the optical properties. If optical properties 130 are taken from a scattering database, then the density and integration size range are instead implied by the settings for that database. Any placeholder type can be associated with any hydrometeor from the databases. The number of hydrometeor types in the lookup tables is unlimited, so the placeholder types can be re-used multiple times. Their names should not be taken too literally, such as in the use of the "snow" placeholder type with large-scale snow, and the "graupel" placeholder for frozen

**Table 2.** Placeholder hydrometeor types

| Name | Density [kg m$^{-3}$] | $D_{min}$ [m] | $D_{max}$ [m] | Frozen |
|---|---|---|---|---|
| Rain | 1000 | $1\times10^{-4}$ | $1\times10^{-2}$ | F |
| Snow | 100 | $1\times10^{-4}$ | $2\times10^{-2}$ | T |
| Graupel | 400 | $5\times10^{-4}$ | $3\times10^{-3}$ | T |
| Aggregate | 900 | $5\times10^{-3}$ | $2\times10^{-2}$ | T |
| Cloud water | 1000 | $5\times10^{-6}$ | $1\times10^{-4}$ | F |
| Cloud ice | 900 | $5\times10^{-6}$ | $1\times10^{-4}$ | T |
| Totalice | 500 | $5\times10^{-6}$ | $2\times10^{-2}$ | T |

T=true; F=false

particles in convective clouds. The "totalice" placeholder was designed for the Met Office global model, which represents all
ice particles as a single species (e.g. Doherty et al., 2007).

To save time, the table generator first calculates optical properties for the full set of possible channels (currently 136) and
then it composes these channels into sets corresponding to satellite instruments (where currently 34 are represented). The list of
channels and instruments is defined in a configuration file, along with the chosen hydrometeor types and physical assumptions
as summarised in Table 1. One output file is then produced per instrument, containing the optical properties for each channel,
tabulated as a function of water content, temperature and hydrometeor type. The relevant grids are fixed at:

- For water content, 401 logarithmically spaced points from $1\times10^{-6}$ kg m$^{-3}$ to $1\times10^{-2}$ kg m$^{-3}$.

- For temperature, 70 points, one every Kelvin, from 204 K to 273 K for frozen hydrometeors and from 234 K to 303 K for
  liquid hydrometeors (frozen or liquid is defined according to Table 2).

The channels for any target instrument can be specified in one of two ways. The normal way is the "condensed" approach,
designed for unpolarised optical properties. Instruments such as conical-scanning microwave imagers may separately measure
vertical and horizontal polarisations at one frequency, but in the condensed representation this is represented as one channel in
the tables. This eliminates redundancy in the data file and leaves it up to the radiative transfer model to remap the condensed set
of channels onto the actual channel set of the instrument. An alternative "full" option is for use with polarised optical properties
(Sec. 3.3) and in this case, the full channel set, including each distinct polarisation, is represented in the data file.

Given that over the width of a channel, bulk scattering properties vary slowly with frequency, there is no attempt to represent
the exact spectral response function. Instead, channels are specified by the central frequency and optical properties are evaluated
at this exact frequency. In the case of double-sideband channels such as $183\pm7$ GHz, the calculations are done at each of the
two frequencies, e.g. 174 GHz and 190 GHz, and then averaged.

Within the main RTTOV-SCATT radiative transfer code, the atmospheric profile is specified, at each model level, in terms of
variables including temperature, humidity, and the sub-grid fraction and grid-box average mixing ratio of each of the hydrom-





eteors represented in the hydrotables. The hydrometeor bulk optical properties are retrieved from the hydrotables as a function of temperature, in-cloud water content, and channel. These are then summed over the set of hydrometeors, together with the gas optical properties driven mainly by oxygen and water vapour absorption, to provide the total bulk optical properties of each layer in the model (see e.g. Bauer, 2001). These profiles are then input to the solver for scattering radiative transfer, which in

the case of RTTOV-SCATT uses a delta-Eddington approach (for further details, see Bauer et al., 2006). Sub-grid heterogeneity of cloud fields is represented through an effective cloud fraction (Geer et al., 2009).

### 2.1 Single-particle optical properties

Single-particle optical properties are derived using either Mie theory, which assumes spherical particles, or from scattering databases that summarise the properties of non-spherical particle habits, which have been computed using more sophisticated

methods such as DDA. Currently, the Liu (2008) and ARTS (Eriksson et al., 2018) databases are available within the RTTOV-SCATT hydrotable generator.

#### 2.1.1 Mie spheres

The Mie theory is solved using an iterative method that computes a set number of terms from the infinite Mie series, using recursion relations to evaluate the required polynomials (see e.g. Ulaby et al., 1981). These calculations depend only on the

size parameter $x_{\mathrm{g}} = \pi D_{\mathrm{g}}/\lambda$ (where $D_{\mathrm{g}}$ is the diameter of the sphere and $\lambda$ is the wavelength) and the complex refractive index of the material composing the sphere, $n = \sqrt{\epsilon}$, where $\epsilon$ is the complex permittivity. For spheres composed of liquid water the permittivity models of Liebe (1989), Kneifel et al. (2014) and Rosenkranz (2015) are available. These models were evaluated in a weather forecasting context by Lonitz and Geer (2019). The Liebe model was the original option, but it is now known to have unrealistic behaviour at low temperatures and is retained only to allow backward evaluation. The Kneifel et al. and

Rosenkranz models are based on recent permittivity measurements and gave better performance, with improved fit between forecast model and observations from SSMIS and other microwave imagers in areas of supercooled liquid water cloud at high latitudes. However, the Kneifel et al. model is only valid up to $500\,\mathrm{GHz}$ so the default and recommended option is the Rosenkranz model, which covers the full microwave and sub-mm range.

Frozen hydrometeors can be modelled as Mie spheres but this is not recommended except for the smallest particles, where the

scattering is in the Rayleigh regime ($x_{\mathrm{g}} << 1$; see Sect. 3.2.4). The Mie representation of frozen hydrometeors gives excessive scattering brightness temperature depressions at lower microwave frequencies, but it generates insufficient scattering at higher frequencies, failing to reproduce the observed behaviour (Geer and Baordo, 2014). This is primarily due to excessive forward scattering from the idealised spherical particle (Sect. 4). However, the Mie capability is still used, where appropriate, to fill gaps in the scattering databases. Specifically, this supports an optional extension of the size ranges below the smallest available

particles from the databases. It is also used to fill the gap below $3\,\mathrm{GHz}$ where the Liu (2008) database does not provide data. Frozen Mie spheres are assumed to be composed of a mixture of air and ice at the relevant density from Table 2, with pure ice assumed to have a density of $917\,\mathrm{kg\,m^{-3}}$ and air a density of $1.225\,\mathrm{kg\,m^{-3}}$. Alternatively, formulations of density as a function of particle diameter can be used, from Wilson and Ballard (1999), Jones (1995) or Brown and Francis (1995). These options





were explored by Doherty et al. (2007). Permittivity of ice uses the Mätzler (2006) formulation, consistent with the ARTS
scattering database (Eriksson et al., 2018). Earlier versions of the table generator followed the Mätzler and Wegmüller (1987)
formulation, which differs only slightly; the option is retained in case exact backward comparison is required. The permittivity
of the ice-air mixture is computed using the Fabry and Szyrmer (1999) mixing rule.

### 2.1.2    Mie-based melting layer

A final Mie-based option, also deprecated, is the melting layer formulation of Bauer (2001), which represents these particles
as a soft ice sphere encased in a layer of water. When this option is selected, and only for nominally frozen hydrometeors, the
resulting estimates of melting particle optical properties are placed in the 273 K temperature bin of the lookup tables. Melting
particles can increase microwave brightness temperatures by 2 to 8 K over radiatively cold surfaces, mainly at frequencies
of 37 GHz and below (Bauer, 2001). The equivalent bright band effect would be important for simulating radar reflectivity.
However, DDA calculations from partially melted ice aggregates show that sphere-based models perform poorly (Johnson
et al., 2016). The representation of non-spherical melting particles is a matter of ongoing research. Hence, melting particles
and bright band effects are not represented by default in the hydrotable generator; this awaits the availability of realistic non-
spherical melting particles in scattering databases.

### 2.1.3    Liu (2008) non-spherical frozen particles

Use of the Liu (2008) scattering database for non-spherical ice particles revolutionised the quality of microwave scattering
simulations made by RTTOV-SCATT (Geer and Baordo, 2014). Table 3 lists the options, of which the sector snowflake was the
previous default choice for snow. The other options are a dendrite snowflake and a variety of hex-plates, columns and rosettes.
These habits are geometric models of ice particles which are rescaled and then discretised into a 3D grid of polarizable points
for input to DDA scattering calculations. To create the database, single-particle scattering properties have been averaged over
a large number of random orientations, and these averages have been tabulated at a range of particle sizes (Liu, 2008, their
Table 2), frequencies from 3 to 340 GHz and temperatures from 233.15 to 273.15 K. The database offers linear interpolation
and extrapolation to a specified frequency, temperature and size. However, this extrapolation is not used within the hydrotable
generator: For frequencies below 3 GHz, Mie sphere results are substituted; above 340 GHz, use of the Liu database is forbid-
den. For temperatures below 233.15 K the optical properties at 233.15 K are substituted. Finally, the available particle sizes
define the size range over which the PSD is integrated. This integration range is shown in Table 3, with $D_{\text{max}}$ being the largest
size available from the database and $D_{\text{min}}$ the smallest, but bounded at 100 $\mu$m on the assumption that the Field et al. (2007)
PSD will be used (see Sect. 3.1).

    The geometric particle model also provides the link between the particle size (the geometric diameter or maximum dimension
$D_{\text{g}}$) and its mass $m$, via the mass-size relation:

$$m = aD_{\text{g}}^{b} \tag{5}$$



**Table 3.** Particles available from the Liu (2008) and ARTS (Eriksson et al., 2018) databases

| ID | Name | $D_{\min}$ [m] | $D_{\max}$ [m] | a | b |
|----|------|--------|--------|---|---|
| | Liu | | | | |
| 0 | Long hex. column | $1.50\times10^{-4}$ | $4.80\times10^{-3}$ | 37.09 | 3.00 |
| 1 | Short hex. column | $1.00\times10^{-4}$ | $3.30\times10^{-3}$ | 116.1 | 3.00 |
| 2 | Block hex. column | $1.00\times10^{-4}$ | $2.50\times10^{-3}$ | 229.7 | 3.00 |
| 3 | Thick hex. plate | $1.00\times10^{-4}$ | $3.20\times10^{-3}$ | 122.7 | 3.00 |
| 4 | Thin hex. plate | $1.50\times10^{-4}$ | $5.00\times10^{-3}$ | 32.4 | 3.00 |
| 5 | 3-bullet rosette | $1.00\times10^{-4}$ | $1.00\times10^{-2}$ | 0.32 | 2.37 |
| 6 | 4-bullet rosette | $1.00\times10^{-4}$ | $1.00\times10^{-2}$ | 0.06 | 2.12 |
| 7 | 5-bullet rosette | $1.00\times10^{-4}$ | $1.00\times10^{-2}$ | 0.07 | 2.12 |
| 8 | 6-bullet rosette | $1.00\times10^{-4}$ | $1.00\times10^{-2}$ | 0.09 | 2.13 |
| 9 | Sector snowflake | $1.00\times10^{-4}$ | $1.00\times10^{-2}$ | 0.002 | 1.58 |
| 10 | Dendrite snowflake | $1.00\times10^{-4}$ | $1.00\times10^{-2}$ | 0.01 | 1.90 |
| | ARTS | | | | |
| 1 | Plate type 1 | $1.32\times10^{-5}$ | $1.00\times10^{-2}$ | 0.76 | 2.48 |
| 2 | Column type 1 | $1.44\times10^{-5}$ | $1.00\times10^{-2}$ | 0.038 | 2.05 |
| 3 | 6-bullet rosette | $1.56\times10^{-5}$ | $1.00\times10^{-2}$ | 0.49 | 2.43 |
| 4 | Perpendicular 4-bullet rosette | $1.80\times10^{-5}$ | $1.00\times10^{-2}$ | 0.32 | 2.43 |
| 5 | Flat 3-bullet rosette | $1.99\times10^{-5}$ | $1.00\times10^{-2}$ | 0.24 | 2.43 |
| 6 | ICON cloud ice | $1.29\times10^{-5}$ | $1.00\times10^{-2}$ | 1.59 | 2.56 |
| 7 | Sector snowflake | $2.00\times10^{-5}$ | $1.02\times10^{-2}$ | 0.00082 | 1.44 |
| 8 | Evans snow aggregate | $3.20\times10^{-5}$ | $1.18\times10^{-2}$ | 0.20 | 2.39 |
| 9 | 8-column aggregate | $1.94\times10^{-5}$ | $9.71\times10^{-3}$ | 65.4 | 3.00 |
| 10 | Large plate aggregate* | $1.62\times10^{-5}$ | $2.29\times10^{-2}$ | 0.21 | 2.26 |
| 11 | Large column aggregate* | $2.42\times10^{-5}$ | $2.00\times10^{-2}$ | 0.28 | 2.44 |
| 12 | Large block aggregate* | $1.32\times10^{-5}$ | $2.19\times10^{-2}$ | 0.35 | 2.27 |
| 13 | ICON snow* | $1.65\times10^{-5}$ | $2.00\times10^{-2}$ | 0.03 | 1.95 |
| 14 | ICON hail* | $1.03\times10^{-5}$ | $5.35\times10^{-3}$ | 383.5 | 2.99 |
| 15 | Gem graupel* | $1.94\times10^{-5}$ | $6.60\times10^{-3}$ | 172.8 | 2.96 |
| 16 | Liquid sphere | $1.24\times10^{-6}$ | $5.00\times10^{-2}$ | 523.6 | 3.00 |

Coefficients a and b describe the mass-size relation $m = aD_g^b$ and are in SI units; see the code for full numerical precision. ARTS IDs are unique to RTTOV and do not correspond to Eriksson et al. (2018); Liu IDs do correspond to Liu (2008). *ARTS standard habits with IDs from 10 to 15 are a mixture of two habits, with the small size range covered by thick plate, long column, block column, short column, gem cloud ice and 8-column aggregate respectively.





There is some ambiguity in the fitting of these coefficients to a particle model (Geer and Baordo, 2014). In this work, $a$ and $b$ coefficients appropriate to the Liu particle models have been taken from Kulie et al. (2010, their Table 1). This choice means that some particles are affected by a slightly unrealistic choice of ice density (Geer and Baordo, 2014), but these values have been retained with the aim of consistency with earlier results.

The optical properties of the Liu particles are illustrated later in Fig. 9 (or see also Kulie et al., 2010; Geer and Baordo,
2014). A limitation of the Liu database is the relatively low diversity among the bulk optical properties achievable using the different habits. For example, the 4-, 5- and 6-bullet rosettes give similar results to the sector snowflake. Then, there is a big gap to the next-most scattering particle, the 3-bullet rosette, and another big gap to the intensely scattering hex. plates and columns which provide similar results as each other. However, these five hex. particles with $b = 3$ provide a uniquely strong bulk extinction that cannot be obtained from the ARTS database (further discussion in Sect. 4).

**2.1.4 ARTS database**

The ARTS scattering database (Eriksson et al., 2018) was created to support sub-mm frequencies and to provide a broader range of non-spherical ice particles, including a variety of aggregates and densely rimed particles (e.g. hail and graupel). Table 3 summarises the options available. The current default frozen particles in RTTOV-SCATT are based on ARTS particles (see Table 1).

The ARTS database provides optical properties at 34 frequencies from 1 to 886 GHz, at three temperatures (190, 230, and 270 K), and at least 34 sizes per habit. The ARTS standard habits are used here; these simplify the application of the database by ensuring a full coverage of size, temperature and frequency. The size issue is that in the underlying database, the smallest size of aggregate habits can exceed 200 μm, so where necessary the standard habits consist of a habit mix. In these cases, the small size range is covered by a single crystal habit having similar shape as the constituents of the aggregate (see Table. 3). For
example, the "Large Plate Aggregate" habit is complemented with the "Thick plate" habit. To avoid discontinuities, there is a linear transition between the two habits over a certain size range. These "mixed" standard habits are throughout named after the habit covering the main size range and contain at least 42 particle sizes. Remaining standard habits are essentially a copy of the original ones. To improve temperature coverage in the standard habits, points at 210 and 250 K are added by a second order interpolation, in order to decrease the error by subsequent linear temperature interpolation. Finally, due to limitations in
DDA, there are some gaps in the database for combinations of large size and high frequencies. In the standard habits, these gaps are filled by copying data from lower frequencies that should be a better approximation than setting the values to zero.

When producing the RTTOV-SCATT hydrotables, the standard habit data are interpolated to the required temperature, frequency and particle size using 3D linear interpolation, with linear extrapolation also permitted up to a limit. This is used to provide values outside the available temperature range, but extrapolation in frequency or size is not used. This is because the
frequency range already covers all current instruments, and the size range of the available particles from Table 3 is also used as the integration range $D_{\mathrm{min}}$ to $D_{\mathrm{max}}$. The bulk optical properties of the ARTS particles will be explored in the rest of this work: see in particular Figs. 2, 9, 10 and 12. In most places in this work, the shortest unambiguous name (such as "ARTS plate") is used.



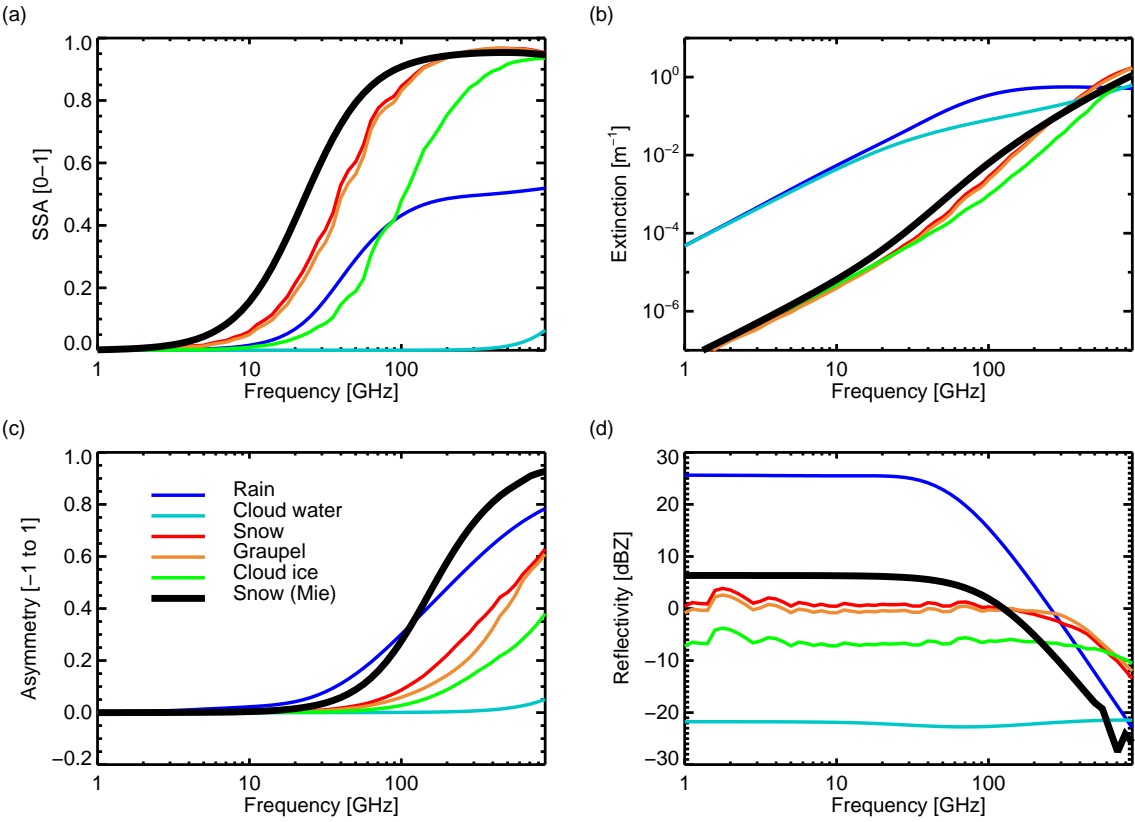

**Figure 1.** Bulk optical properties: (a) Single scattering albedo; (b) Extinction; (c) Asymmetry; (d) Radar reflectivity for the 5 default hydrometeor types in RTTOV-SCATT (Table 1) plus soft Mie spheres representing snow (density 100 kg m$^{-3}$, Field et al. (2007) tropical PSD). Computations have been done for a water content $l = 1 \times 10^{-4}$ kg m$^{-3}$ and temperatures $T$ = 253 K for frozen particles and $T$ = 283 K for liquid particles. Frequency steps are logarithmically distributed with 20 points per decade.

## 2.2 Bulk optical properties

Figure 1 shows the spectral variation of bulk optical properties across the microwave and sub-mm frequencies. These properties have been generated from the default 5-hydrometeor configuration (Table 1) for a water content $l = 1 \times 10^{-4}$ kg m$^{-3}$. A Mie sphere snow particle is also included to support discussions in Sec. 4. Rain and cloud water have broadly similar extinction per mass of particles (panel b), but because the rain particles are large enough to be in the Mie regime across most of the frequency range, they have much increased single scattering albedo (up to 0.5, panel a), asymmetry (up to 0.8, panel b) and

radar reflectivity (up to 27 dBZ, panel d). Cloud water starts to depart from the Rayleigh regime above around 500 GHz, with non-zero values of the single scattering albedo and asymmetry. Moving to snow, graupel and cloud ice, these have generally much lower extinction than the liquid particles below 100 GHz, but this reverses above around 300 GHz. Snow and graupel provide substantial scattering above around 20 GHz (SSA > 0.3), becoming almost purely scattering particles above 150 GHz





(SSA $\simeq 0.95$). Cloud ice has an order of magnitude less extinction than snow and graupel at 100 GHz, and much less scattering
(lower SSA) across the whole frequency range. Compared to snow and graupel, this arises mainly from the choice of PSD, which gives much smaller numbers of large particles. Figure 1 hence shows the "spectral signatures" of hydrometeors and illustrates the utility of making measurements across the whole of the microwave and sub-mm in order to characterise the physical details of cloud and precipitation particles.

The only difference between snow and graupel in the default configuration (Table 1, Fig. 1) is the use of, respectively, an
ARTS large plate aggregate and an ARTS column. The primary resulting difference is the asymmetry, with graupel giving less forward scattering between 50 GHz and 500 GHz. This allows the graupel to generate deeper brightness temperature depressions (see Fig. 8 later). This greater "scattering" ability led to its selection as a reasonable representation of convective snow (Geer, 2021b).

In Fig. 1 the frozen particles have small oscillations with frequency, particularly obvious in the radar reflectivity at lower
frequencies. This is a result of interpolating away from the original temperature, size and frequency steps in the ARTS database. This should not be an issue near the frequencies of typical satellite channels, since the ARTS frequencies have been chosen with this in mind (Eriksson et al., 2018).

Figure 2 illustrates the full range of frozen particle representations available from the ARTS database, along with the Mie sphere. The ARTS particles fall into two classes. The first is less dense particles with branched shapes including rosettes,
snowflakes and most of the aggregates. These are shown with solid lines and typically generate smaller SSA, extinction, asymmetry and radar reflectivity. The second class is denser and more compact particles including pristine crystals, densely rimed particles (graupel and hail) and the 8-column aggregate. These are shown with dashed lines and typically generate higher values of all the optical properties. Further discussion, and comparison to the Liu (2008) particles, is made in terms of brightness temperature in Sec. 4.

**2.2.1 Importance of mass-size relation**

The mass-size relation (Eq. 5, specified by the $a$ and $b$ coefficients from Table 3) plays an important role in controlling the bulk optical properties derived from non-spherical frozen particles. In the microwave and sub-mm, the primary control over the single-particle optical properties is the particle's mass (e.g. Eriksson et al., 2015). Hence the mass-size relation already describes a lot about how particle size (as specified by the assumed PSD) maps onto optical properties. Further, as will be
shown in Sec. 3.1, the mass-size relation also affects the shape of the PSD itself.

To summarise the available mass-size options, Fig. 3 shows those of the ARTS particles, illustrated using the effective particle density:

$$\rho_{\mathrm{e}} = \frac{m(D_{\mathrm{g}})}{(\pi/6)D_{\mathrm{g}}^3} \simeq \frac{aD_{\mathrm{g}}^b}{(\pi/6)D_{\mathrm{g}}^3}. \tag{6}$$

This is the mass of the particle $m(D_{\mathrm{g}})$ divided by the volume of a sphere of diameter $D_{\mathrm{g}}$. For spherical particles, the effective
density and true density are equal. The true particle mass is reported in the particle databases and is the mass of ice used in the DDA calculation; these can vary slightly from the fitted mass-size relation (see e.g. Eriksson et al., 2018, their Fig. 11).



**Figure 2.** Bulk optical properties for the ARTS frozen particles at 183 GHz and 253 K, with the exception of panel d which gives the reflectivity at 94 GHz instead. The Field et al. (2007) tropical PSD has been used in all cases, with the old integration (Sect. 3.2) and a cutoff at $D_{min} = 1 \times 10^{-4}$ m. A soft ice sphere ("Mie") is also included, with settings as Fig. 1. Bulk reflectivity and extinction have been normalised by those of the ARTS large plate aggregate; some lines are off scale to be able to focus on the most populated areas. The legends are in order of brightness temperature depression at 183 GHz, from most to least scattering (see Fig. 9)

Within the hydrotable generator, it is the mass-size relation that is used to estimate the particle mass where required (primarily, in the derivation of the PSD, Sec. 3.1) but this is an approximation. In Fig. 3 the particles with $b$ close to three (the hail, graupel and 8-column aggregate) have almost constant effective density as a function of particle size. Most of the other particles have



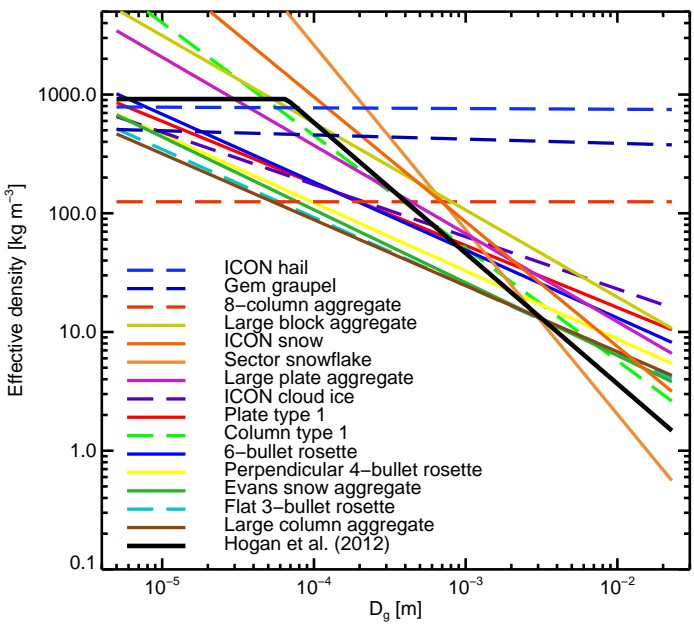

**Figure 3.** Effective density $\rho_e$ implied by the mass-size relations of the particles from the ARTS database, as a function of geometric particle size (maximum dimension $D_g$). Also shown is the Hogan et al. (2012, their Eq. 4) restatement of the Brown and Francis (1995) mass-size relation in SI units and as a function of $D_g$. The ARTS mass-size relations are represented across the whole size range, even if non-physical densities may be generated, since that is how they are used in Eq. 9 (later). The legend is ordered by the effective density at $1 \times 10^{-3}$ m. Fig. 11 of Eriksson et al. (2018) is similar but is based on the reported particle masses, rather than the mass-size relation.

$b$ closer to 2 and hence their effective density decreases strongly with size. Some mass-size relations generate non-physical super-dense particles when taken out of their validity range (as mentioned in Sec. 2.1.1, the assumed density of pure ice in RTTOV-SCATT is 917 kg m$^{-2}$). This is relevant because when the PSD is fitted analytically to the water content (Eq. 9; Sec. 3.1) any superdense region will be included. But as explored in Sects. 3.2.4 and 3.2.2 this is of little practical relevance, and even in the sub-mm the bulk extinction is insensitive to ice particles smaller than 100 $\mu$m.

In an ideal world, users would impose their own constraints on the mass-size relation. For example, a certain mass-size relation may be assumed within the physics of the forecast model which supplies the cloud and precipitation profiles, and it may be intended to achieve microphysical consistency throughout the modelling chain. Further, there are observational constraints, and for example the Brown and Francis (1995) mass-size relation gives a good description of midlatitude stratiform ice cloud (Hogan et al., 2012); this is shown on Fig. 3. However, there is currently no way of decoupling the mass-size relation from

the DDA particle choice in the hydrotable generator; this could only be achieved by choosing an appropriate (and probably different) database particle for each size bin - in other words, a particle ensemble approach, which is not yet supported. The Mie sphere does allow a free choice of mass-size relation (Sec. 2.1.1) but obviously brings many other drawbacks, so this is not advised. However, Fig. 3 shows that the available DDA particles span a wide range of mass-size possibilities; the





dendrite particle in the Liu database (Table 3) would almost exactly match the Brown and Francis (1995) mass-size relation,
for example. However, microwave radiances have their strongest sensitivity to convective snow particles, both in the tropics
and midlatitudes, so the appropriate mass-size relation remains poorly known. Hence the dominant approach is to explore all
potential DDA particle choices and to use the one that provides the best fit between model and observations (e.g. Geer, 2021b,
and references therein). Interestingly, the best choices in that work, reflected in the default RTTOV-SCATT configuration
(Table 1) seem to be particles with $b$ around 2.0 - 2.4; particles with $b$ closer to 3 seem to work poorly as a description of
convective snow.

The sensitivity of optical properties to the mass-size relation is further illustrated by the sector snowflake, which is present
in both ARTS and Liu (2008) databases and has almost identical optical properties as a function of particle size. However, as
shown in Table 3, the $a$ and $b$ coefficients used with the Liu and ARTS databases are different, due to different, but equally valid
methodological choices in fitting those coefficients to the particle masses within the databases (see Geer and Baordo, 2014,
appendix B for further explanation). These small differences still have a significant effect on the bulk optical properties. The
ARTS sector snowflake provides less scattering than the Liu equivalent, and simulations of very thick clouds using the ARTS
sector snowflake and the Field et al. (2007) PSD can be up to 20 K warmer around 300 GHz (shown in Fig. 9 later). Using an
identical $a$ and $b$ it is possible to eliminate this difference. However, it is preferred to retain the values previously used with the
Liu database to ensure full back-reproducibility, but for the ARTS database to use the coefficients that are supplied with that
database.

Further discussion on the importance of the mass-size relation is found in Sects. 3.1 and 3.2.4. In this work, it is important
to realise that when the bulk optical properties of a particle habit are discussed, this is the net result of both the physical
characteristics of the individual particles and the effect of the corresponding mass-size relation on the PSD.

## 3 Methods in detail

### 3.1 Particle size distributions

The table generator was revised at version 13.0 for a more flexible handling of PSDs and a wider set of options. Underlying
most of the PSDs is the Modified Gamma Distribution (MGD):

$$n_{\mathrm{g}}(D_{\mathrm{g}}) = N_0 D_{\mathrm{g}}^{\mu} \exp\left(-\Lambda D_{\mathrm{g}}^{\gamma}\right). \tag{7}$$

This follows the universal framework of Petty and Huang (2011). The version of the MGD used here is based in geometric
diameter or maximum dimension, $D_{\mathrm{g}}$, consistent with the majority of available PSD formulations. $n_{\mathrm{g}}(D_{\mathrm{g}})$ is the number
density of particles per unit of particle diameter, e.g. [m$^{-3}$ m$^{-1}$] or simply [m$^{-4}$]. $N_0$, $\mu$, $\Lambda$ and $\gamma$ are the four parameters of
the MGD; the units of $N_0$ and $\Lambda$ are dependent on the units of the particle size descriptor (e.g, $D_{\mathrm{g}}$ in m) and the values of $\mu$
and $\gamma$, which are themselves dimensionless.





The moments of the MGD, $M_k$, can be derived analytically (Petty and Huang, 2011):

$$M_k := \int_0^\infty D_g^k n_g(D_g)\, dD_g = \frac{N_0}{\gamma} \frac{\Gamma\left(\frac{\mu+k+1}{\gamma}\right)}{\Lambda^{(\mu+k+1)/\gamma}}, \tag{8}$$

where the Gamma function $\Gamma(z) = \int_0^\infty x^{z-1}\exp(x)\,dx$ arises naturally from the integration of the MGD. This is computed in the table generator by means of a built-in Fortran function.

The PSD is fitted to the hydrometeor water content $l$. Where a power law mass-size relation is known, and $a$ and $b$ are its coefficients (Eq: 5), $l$ is proportional to the $b$th moment of the PSD:

$$l = \int_0^\infty a D_g^b n_g(D_g)\, dD_g = aM_b. \tag{9}$$

Typically all but one parameter of the MGD is prescribed and the remaining "free parameter" is adjusted to fit the hydrometeor water content. The table generator allows either $N_0$ or $\Lambda$ to be the free parameter since the other two are less mathematically convenient. These are hence computed from Eqs. 8 and 9 as follows:

$$N_0 = \frac{l\gamma\Lambda^p}{a\Gamma(p)} \tag{10}$$

or

$$\Lambda = \left(\frac{aN_0\Gamma(p)}{l\gamma}\right)^{\frac{1}{p}} \tag{11}$$

where $p = (\mu+b+1)/\gamma$. There are a couple of issues with the analytical approach: first, any numerical integration of the PSD, such as to obtain the bulk optical properties, is necessarily done over a limited size range $D_{min}$ to $D_{max}$ (see Eq. 1); second, some particles with $b < 3$ in the mass-size relation can generate non-physical super-dense small particles (Sect. 2.2.1 and Fig. 3). In the hydrotable generator these are partially dealt with via "renormalisation", an empirical rescaling of the PSD described in Sec. 3.2.2.

The PSDs available in the table generator are summarised in Table 4. The Marshall and Palmer (1948, MP48) PSD is used for rain in the default configuration and is also a possibility for snow. This PSD has $\mu = 0$ and $\gamma = 1$, producing what is classed as an exponential distribution; in the table generator, fixed values of $N_0$ are specified for liquid or frozen hydrometeors and $\Lambda$ is the free parameter (see Table 4). It is optionally possible to add a temperature dependent $N_0$ (Panegrossi et al., 1998, appendix) which represents the collection of smaller droplets by larger drops during sedimentation.

The "gamma" distribution, where $\gamma = 1$, is often used for cloud water or cloud ice. Typically $\mu$ and $\Lambda$ are prescribed and $N_0$ becomes the free parameter. The default configuration for cloud water follows this approach with fixed parameters that ensure cloud water particles are in the Rayleigh regime at microwave frequencies (see Table 4). The pre-v13 equivalent is retained for back-comparison purposes; this used an alternative power law fit to Eq. 10, but the resulting difference in bulk optical properties is minimal. Prior to v13.0, the gamma distribution was also used for cloud ice, also with an alternative formulation for Eq. 10. An MGD implementation is examined later in this section, labelled "MGD A". However, the default PSD for cloud





**Table 4.** Available PSDs

| Abbreviation | Name | MGD parameters | | | |
| --- | --- | --- | --- | --- | --- |
| | | $N_0$ | $\mu$ | $\Lambda$ | $\gamma$ |
| MGD | Modified Gamma Distribution | user specified | | | |
| MP48 (rain) | Marshall and Palmer (1948) | $8\times10^6$ | 0.0 | free | 1.0 |
| MP48 (snow) | Marshall and Palmer (1948) | $4\times10^6$ | 0.0 | free | 1.0 |
| Gamma (water) | pre-v13 gamma (cloud water) | free* | 2.0 | $2.13\times10^5$ | 1.0 |
| MGD W | MGD implementation at v13.0 for cloud water | free | 2.0 | $2.13\times10^5$ | 1.0 |
| Gamma (ice) | pre-v13 gamma (cloud ice) | free* | 2.0 | $2.05\times10^5$ | 1.0 |
| MGD A | MGD implementation of pre-v13 gamma (cloud ice) | free | 2.0 | $2.05\times10^5$ | 1.0 |
| MGD B | MGD implementation of v13.0 cloud ice | free | 0 | $1.0\times10^4$ | 1.0 |
| H13 | Heymsfield et al. (2013) for frozen particles | free | parametrised | | 1.0 |
| F05 | Field et al. (2005) for frozen particles | not applicable | | | |
| F07 | Field et al. (2007) for frozen particles | not applicable | | | |
| MH97 | McFarquhar and Heymsfield (1997) for frozen particles | not applicable | | | |

* here, the free parameter is set using an alternative method to Eq. 10 – see text.

ice at v13.0 ("MGD B") was identified by parameter estimation (Geer, 2021b) and is similar to the Heymsfield et al. (2013) PSD where its distribution becomes close to exponential ($\mu = 0$).

The Heymsfield et al. (2013, H13) cloud ice parametrisation prescribes various temperature-dependent functions for $\mu$ and $\Lambda$; these are based on aircraft measurements of ice cloud from the Arctic to the Tropics. Three of the H13 configurations are available in the table generator: the stratiform (M), convective (C) or "all" (A) approach (which takes their "composite" form for $\Lambda$). The PSDs of Field et al. (2005, F05), Field et al. (2007, F07) and McFarquhar and Heymsfield (1997, MH97) are implemented outside the MGD framework and their additional details are covered in following subsections.

Figure 4a explores typical options for snow and graupel particles, using the default snow particle, the ARTS large plate aggregate. This has a mass-size relation with $b = 2.26$, within the range of typical choices for snow and aggregates. Geer and Baordo (2014) rejected MP48 in favour of the F07 tropical (T) PSD, in order to reduce numbers of the very largest particles, which were producing too much scattering. Geer (2021b) confirmed F07 T as a reasonable choice for both large-scale and convective snow ("graupel"); it is now the default option for both. The F07 midlatitude (M) and the Field et al. (2005) PSDs

are also available and could help further reduce the number of very large snow or graupel particles if needed.

Figure 4b explores possible PSDs for cloud ice. The pre-v13 gamma distribution (labelled MGD A, see Table 4) made the particles very small and the simulated ice cloud was almost invisible at frequencies of 183 GHz and below. Geer (2021b) explored other options, hoping to make cloud ice more visible, as seen in observations (e.g. Doherty et al., 2007; Hong et al., 2005). A number of aircraft-based PSDs were tested (H13 S, F07 M, MH97) but all produced too much scattering, even when



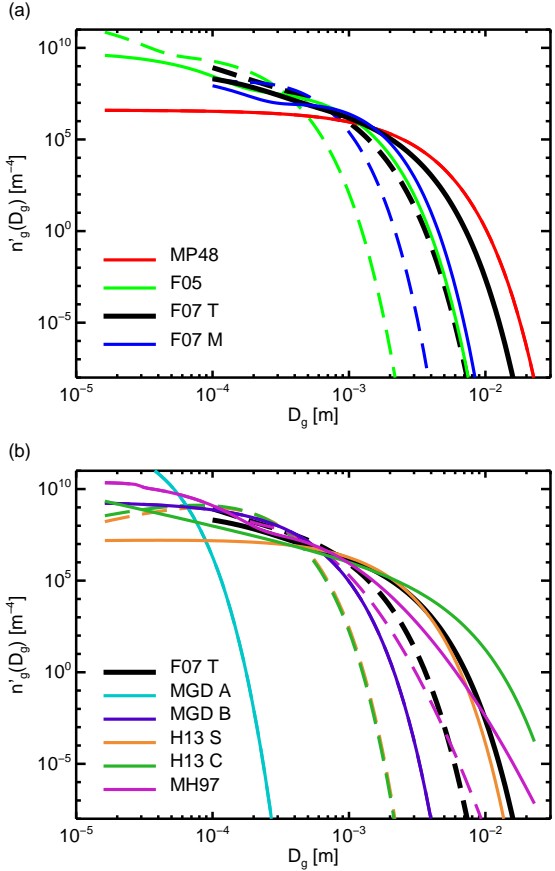

**Figure 4.** Examples of PSDs for frozen hydrometeors available in RTTOV-SCATT, using an ARTS large plate aggregate ($m = 0.21 D_{\mathrm{g}}^{2.26}$ [SI]), an ice water content of $1 \times 10^4$ kg m$^{-3}$, and temperatures of 223 K (dashed) and 263 K (solid). PSDs have been split across two panels for clarity but for reference F07 T is present in both. At 223 K, the H13 S and H13 C distributions are nearly identical below $n_{\mathrm{g}}'(D_{\mathrm{g}}) = 10^7 \mathrm{m}^{-4}$. See Table 4 for PSD name abbreviations. PSDs have been renormalised (Sec. 3.2.2) and all have been integrated using the new approach (Sec. 3.2).

the particle type was chosen to generate as little scattering as possible (e.g. the ARTS large column aggregate). Figure 4 shows that these PSDs can generate significant numbers of larger particles, particularly at warmer temperatures, which must be the cause of the excess scattering. To fill the gap between the previous gamma configuration (e.g. MGD A, too few large particles) and the aircraft-based PSDs (too many large particles), new PSDs were created, inspired by the low-temperature part of H13. The configuration labelled here as MGD B (see Table 4) was ultimately chosen as the cloud ice default. Once sub-mm data
from ICI is available, it will be seen whether this is indeed a physically reasonable choice; however it was shown to do a reasonable job in representing observations at 183 GHz.





An issue with many ice PSDs, and particularly evident with F07 and MH97 on Fig. 4, is the presence of a "small mode" of ice particles. The aircraft measurements on which these PSDs were based were subject to probe shattering (Korolev et al., 2011) and optical effects (O'Shea et al., 2020) that, it is now thought, create a spuriously large number of small particles. Hence the small-size mode of these distributions might be non-physical. The H13 PSD is intended to be free from at least the probe shattering effect (Heymsfield et al., 2013).

### 3.1.1 Field-type PSDs

The Field et al. (2005, 2007) PSDs are based on a "universal" rescaled PSD $\Phi_{23}(x_{23})$, which is a function of a non-dimensional particle size parameter $x_{23} = D_{\mathrm{g}} \frac{M_2}{M_3}$. Here, $M_2$ and $M_3$ are the second and third moments of the PSD (Eq. 8) but any pair of moments could have been used. The universal PSD is parametrised as the sum of exponential and gamma PSDs in $x_{23}$ which gives the resulting PSD a characteristic population bulge in the smaller sizes (Fig. 4).

The size-based PSD is recovered by

$$n_{\mathrm{g}}(D_{\mathrm{g}}) = M_2^4 M_3^{-3} \Phi_{23}(x_{23}). \tag{12}$$

To evaluate the PSD hence requires knowledge of $M_2$ and $M_3$, or equivalently any other pair of moments; these are obtained by an empirical relation that converts one moment to any other (e.g. Eq. 3 in Field et al., 2007). The water content $l$ provides $M_{\mathrm{b}}$ through Eq. 9; this is first converted to $M_2$ and then $M_2$ is used to obtain $M_3$. The universal PSD is not itself temperature dependent, but Field et al. (2007) provides two parameterisations, one for tropical and one for midlatitude conditions. The temperature dependence arises through the empirical relation between moments, so that the F05 and F07 PSDs generate smaller particles at lower temperatures (Fig. 4).

There are some issues to consider with the Field PSDs, as well as just the small-particle mode noted before. First, the aircraft observations on which they were based did not measure particles with $D_{\mathrm{g}}$ smaller than $1 \times 10^{-4}$ m (100 $\mu$m). The universal PSD can be used to extrapolate to smaller sizes; the hydrotable generator allows this for the F05 PSD. Field et al. (2007) recommended more strongly not to extrapolate, so the table generator terminates the F07 PSD at $D_{\min} = 1 \times 10^{-4}$ m (Fig. 4). When the above procedure is followed to define a size-based PSD from the ice water content, it is assumed that it is valid with an integration over sizes from 100 $\mu$m to infinity. The numerical integration of the resulting $n_{\mathrm{g}}(D_{\mathrm{g}})$ and particle mass $m(D_{\mathrm{g}})$ (following Eq. 9) should recover the original ice water content, but instead the results can be very different; this is covered in Sec. 3.2.2. The F07 T PSD has proved useful in fitting real observations and it is vital to the default configuration of the table generator; however users need to be aware of these complex issues.

### 3.1.2 McFarquhar and Heymsfield (1997) PSD

The McFarquhar and Heymsfield (1997) PSD is based on mass-equivalent diameters $D_{\mathrm{e}}$ where

$$D_{\mathrm{e}} = \left( \frac{6m}{\pi \rho_{\mathrm{ice}}} \right)^{1/3}. \tag{13}$$





Here $m$ is the mass of the particle and $\rho_{\text{ice}}$ is the density of solid ice (note the table generator uses $\rho_{\text{ice}} = 917\,\text{kg}\,\text{m}^{-3}$ compared to $\rho_{\text{ice}} = 910\,\text{kg}\,\text{m}^{-3}$ in the original work). Similar to the Field PSDs, it has two modes (Fig. 4b): the first represents particles smaller than $D_{\text{g}} = 1 \times 10^{-4}$ m (100 $\mu$m) using a gamma distribution in $D_{\text{e}}$. Larger particles are represented by a lognormal

distribution, also in $D_{\text{e}}$; this cannot be represented in the universal MGD framework of Petty and Huang (2011). There is no hard cutoff between the distributions, rather they are summed for all $D_{\text{e}}$ from 0 to $\infty$, observing that the two distributions do not have a big overlap. To adapt the PSDs to the specified ice water content $l$, the water content is first split into two parts representing the small and large particles. This is done based on an empirical relation (McFarquhar and Heymsfield, 1997, Eq. 5). The small- and large-particle PSDs are then dependent on the partial masses (their Eqs. 3 and 4). The parameters of both

PSDs are also temperature dependent (their Eqs. 7-12), producing behaviour broadly similar to Field et al. (2007, see Fig. 4b). A PSD based on geometric diameter is recovered by the conversion

$$n_{\text{g}}(D_{\text{g}}) = \frac{dD_{\text{e}}}{dD_{\text{g}}} n_{\text{e}}(D_{\text{e}}), \tag{14}$$

where $\frac{dD_{\text{e}}}{dD_{\text{g}}}$ has been evaluated numerically.

The MH97 PSD is less sensitive to the choice of mass-size relation and hence less sensitive to variations in the particle habit

(see Fig. 12). This is not, it is thought, because it is based in mass-equivalent diameter $D_{\text{e}}$, as hypothesised by Eriksson et al. (2015), but because it puts so much of the mass in the small particle mode (Ekelund et al., 2020b). As with the Field PSDs, this small-particle mode may be physically incorrect and may have been generated by probe-shattering or optical effects (Korolev et al., 2011; O'Shea et al., 2020).

### 3.2 Integration methods

The core of the hydrotable generator is the numerical integration over the PSD to produce the bulk optical properties, as described earlier (Eqs. 1 to 4). This section first describes the more technical aspects of the integration: numerical integration methods, renormalisation, and diagnostics (Sects. 3.2.1, 3.2.2, and 3.2.3 respectively). Then Sect. 3.2.4 explores the scientific importance of the integration, illustrating the size range of particles that contribute to the bulk optical properties, and helping to explain the impact of different microphysical choices.

### 3.2.1 Numerical integration

In previous versions, numerical integration was done at fixed steps in particle size $D_{\text{g}}$, using a rectangle rule integration, centred on the integration point. The current version also offers an improved integration using the trapezium rule, and with log-spaced integration points in $D_{\text{g}}$ to better resolve the small size ranges. The number of integration points is fixed at 100 and is the same in both methods. The integrations in Eqs. 1 to 4 use a PSD that has been renormalised to conserve integrated mass, $n'_{\text{g}}(D_{\text{g}})$; this

is described in Sect. 3.2.2. For reasons to be explained, a mix of the old and new integration techniques is used in the default configuration (Table 1).

The integration of optical properties is done over the truncated range $D_{\text{min}}$ to $D_{\text{max}}$. For Mie spheres, the integration range is given in Table 2. For particles from the Liu or ARTS databases, the integration size range is taken from Table 3, with



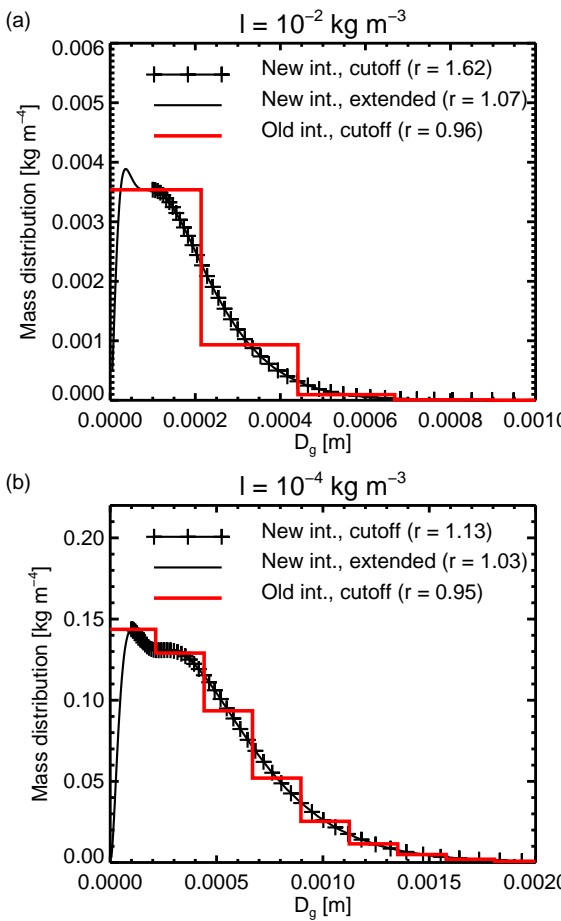

**Figure 5.** Numerical integration options illustrated using the reconstruction of the implied water content (Eq. 15), for the F07 T PSD with the ARTS large plate aggregate and $T = 223$ K. The mass distribution is $a(D_g)^b n_g(D_g)$, using the PSD before renormalisation. The specified water content is (a) $l = 1 \times 10^{-6}$ kg m$^{-3}$ and (b) $l = 1 \times 10^{-4}$ kg m$^{-3}$. The resulting renormalisation factor ($r$, see Sect. 3.2.2) is indicated in the legend. The integration options are the new integration with $D_{min} = 1 \times 10^{-4}$ m (100 $\mu$m) cutoff (black crosses), the new integration extended down to $D_g = 5 \times 10^{-6}$ m (black line), or the old integration (red line).

two exceptions. First is that the size range can optionally be extended down to the relevant $D_{min}$ from Table 2. If selected,
the relevant optical properties are computed using Mie theory, assuming this is valid for particles smaller than the minimum particle size (see Sect. 2.1). If that option is not selected, then a minimum size $D_{min} = 1 \times 10^{-4}$ m is applied when the F07 PSD is used with the ARTS shapes, to avoid extrapolating the PSD. Note that as described in Sect. 2.1, for the implementation of the Liu database in the hydrotable generator, the $D_{min} = 1 \times 10^{-4}$ m constraint was imposed unilaterally in Table 3, meaning that it affects the Liu particles no matter which PSD is chosen (this behaviour is undesirable but is preserved for back-compatibility).
The numerical integration methods are illustrated in Fig. 5 using the computation of the implied water content (Eq. 15, next section). The old method is represented by the stepped red line; its grid was too coarse to resolve sharp PSD features in the





small size ranges. The trapezium rule used in the new method is represented by straight lines between log-spaced integration points (indicated by the black crosses in the example with the $D_{\text{min}} = 1 \times 10^{-4}$ m (100 $\mu$m) cutoff). The new method is a more exact representation of the integration range $D_{\text{min}}$ to $D_{\text{max}}$, whereas in the old method the centred bins extended all the way

down to $D_{\text{g}} = 0$, indeed fractionally beyond in some cases. This means that with the old method, even if the nominal $D_{\text{min}}$ was significantly above zero, the integration was still roughly representing the full size range of particles from zero to infinity. Hence if the intention were to exclude the smallest particles from the PSD, such as when the 100 $\mu$m cutoff is used with the F07 PSD, then the old integration scheme does not fully achieve this aim. The importance of this is examined in the next section.

### 3.2.2   Renormalisation

An important test of the numerical integration is whether the specified water content $l$, used to specify the PSD, can be recovered in the implied water content when the particle mass and PSD are numerically integrated across the chosen integration range $D_{\text{min}}$ to $D_{\text{max}}$:

$$l_{\text{implied}} = \int_{D_{\text{min}}}^{D_{\text{max}}} a D_{\text{g}}^{b} n_{\text{g}}(D_{\text{g}}) \, dD_{\text{g}}. \tag{15}$$

The reconstructed water content may be different due to deficiencies in the numerical integration, if the chosen particle size

range omits part of the PSD, or if there are inaccuracies in the conversion from water content to PSD parameters.

    A renormalisation factor $r$ can be computed:

$$r = \frac{l}{l_{\text{implied}}}. \tag{16}$$

In order not to lose or gain mass, the PSD is renormalised as follows:

$$n_{\text{g}}'(D_{\text{g}}) = r \, n_{\text{g}}(D_{\text{g}}). \tag{17}$$

All the PSDs used in this work are renormalised, with the exception of those shown in Fig. 5. In most cases the renormalisation is minor, with $|r-1|$ less than 0.03, often much smaller. However there are some exceptions. As shown in the figure, the F07 T PSD requires relatively large amounts of renormalisation, with $|r-1|$ of 0.13 and 0.04 in this example (using the new integration with the 100 $\mu$m cutoff, for the cold and warm temperatures respectively). The other main execptions are the MGD A PSD, which has $|r-1| = 0.4$, and the F05 PSD, which has $|r-1| = 9.1$ at the low temperature setting where the problem is

worst (for the same example, but not shown on Fig. 5). Since these are the PSDs with the largest number concentrations in the smallest sizes, this illustrates how the failure of Eq. 15 is typically due to a large amount of mass, and/or sharply peaked PSD features in the very smallest size ranges. This can make numerical integration difficult.

    In the case of the F05 and F07 PSD, the extrapolation of the PSD below $D_{\text{min}} = 1 \times 10^{-4}$ m exposes the question of whether to use this portion of the Field-type PSDs. Comparing panels Figure 5a and b shows that any issues are most relevant for

the smallest water contents. A question is whether to truncate the F07 PSD at $D_{\text{min}} = 1 \times 10^{-4}$ m as recommended (Field et al., 2007), or whether to allow it to extrapolate to smaller sizes. The renormalisation factors are actually largest for the new





integration, with exact truncation at $D_{\min} = 1 \times 10^{-4}$ m. Renormalisations are smaller for the extended integration range and for the old integration, which effectively does not truncate the size distribution. This suggests that the reconstruction of mass using the F07 PSD may be intended to be done with an integration from 0 to infinity. In any case, the issue is that bulk scattering properties and resulting brightness temperatures generated using the F07 PSD can differ markedly depending on these details. However the problem is worse for smaller water contents. Overall, with the F07 PSD, the least renormalisation is generated with the old integration basis (see Sec. 3.2) and with the 100 $\mu$m cutoff; hence even at v13.0 these are the approaches used in the default configuration of hydrometeors based on the F07 PSD. This has the advantage of retaining comparability of the results with earlier work (e.g. Geer and Baordo, 2014). But it is important to realise that the results coming from the F07 PSD are dependent on these choices.

Renormalisation is always active in the table generator, but to alert the user to any significant issues, it will throw an error if the order of magnitude of renormalisation $|\log_{10}(r)|$ exceeds a pre-set threshold. For hydrometeors using the MP48 and MGD PSDs, the thresholds are 0.05 or less, showing they are not much affected. For the hydrometeors using the F07 PSDs, the threshold has to be 0.5. However, the largest renormalisations are for the smallest water contents, meaning the issue does not in most cases have a significant influence on the final simulated brightness temperatures.

### 3.2.3 Diagnostic mode

As illustrated in this subsection, there are many complexities to the apparently simple task of numerical integration of bulk optical properties or mass, particularly since many PSDs put significant mass in the smallest size ranges, where the particles are unimportant in the microwave and sub-mm radiative transfer. Since it has not been possible to demonstrate or test every combination of options provided by the tool, users may wish to check the quality of the integrations for themselves. If the amount of renormalisation required is large, this is an early warning of problems, but even better is to make use of the new diagnostic mode, which writes out an additional diagnostic text file during the generation of the lookup tables. For a chosen particle ID (from Table 2), temperature, frequency, and water content, the diagnostic mode outputs the values of key parameters at each integration point: $D_g$, $D_m(D_g)$, $m(D_g)$, $N'_g(D_g)$, $\beta_e(D_g)$, $\beta_s(D_g)$, $\beta_b(D_g)$, $g_{\text{single}}(D_g)$ and the "extagrand", the numerical representation of $\beta_e N'_g(D_g)dD_g$, which is summed to create the final bulk integrated extinction. The resulting bulk values are also provided, along with the renormalisation factor $r$ to be able to recreate $N_g(D_g)$. The new diagnostic mode was heavily used in the development of v13.0 and in the preparation of figures for this paper.

### 3.2.4 Converting mass to extinction

Figure 6 illustrates the integration of extinction (Eq. 1) for frozen particles at 190.31 GHz using the F07 T PSD. The integration combines the single-particle extinction ($\beta_e(D_g)$, panel a) and the PSD ($N'_g(D_g)$, panel b), so that each integration element gives a contribution of $\beta_e N'_g(D_g)dD_g$ to the bulk extinction (panels c and d), referred to here as the "extagrand". The elements are logarithmic in this example (the "new" integration has been used) and the size axis of the plot is logarithmic; hence the bulk extinction is proportional to the area under the curves in panels c and d. All panels have been normalised: the extinction and the size distribution have been respectively divided and multiplied by the single-particle mass (based on Eq. 5), presenting Eq. 1



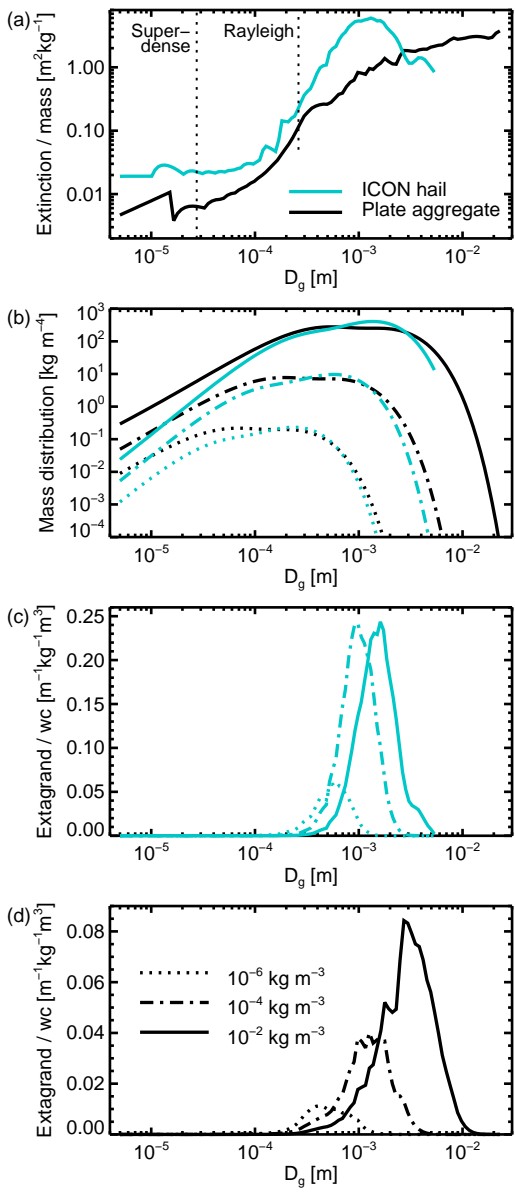

**Figure 6.** Integration of single-particle extinction over the PSD: (a) single-particle extinction per unit particle mass $\beta_e(D_g)/m(D_g)$; (b) mass distribution $m(D_g)N'_g(D_g)$; (c) contribution to total extinction ("extagrand") normalised by water content $\beta_e(D_g)N'_g(D_g)dD_g/l$ for ARTS ICON hail; (d) as (c) but for ARTS plate aggregate. The F07 T PSD has been used with the "new" integration, at a temperature of 253 K and a frequency of 190.31 GHz. The extended integration range for cloud ice (down to $5\times10^{-6}$ m) has been used for illustrative purposes. Dotted lines on (a) correspond to, first, the largest sizes for which the plate aggregate is super-dense; second, the largest size for which Rayleigh scattering would be an appropriate model for the ICON hail particle.





as follows:

$$\beta_{\mathrm{e}} = \int\limits_{D_{\min}}^{D_{\max}} \sigma_{\mathrm{e}}(D_{\mathrm{g}})/m(D_{\mathrm{g}}) \times m(D_{\mathrm{g}})n_{\mathrm{g}}'(D_{\mathrm{g}}) \, dD_{\mathrm{g}}. \tag{18}$$

This has two aims; first to normalise quantities that would otherwise vary over more than 10 orders of magnitude; second and most importantly, to focus on the key process in the computation of bulk optical properties, which is to convert hydrometeor mass to bulk extinction. Further, to more easily compare the results with different water contents in panels c and d, the extagrand

has been normalised by the respective hydrometeor water content. Panel b shows that the effect on the F07 PSD of increasing the hydrometeor water content is not just to increase the overall mass of particles, but also to significantly increase the maximum particle sizes included.

For the ICON hail particle and a water content of $l = 1 \times 10^{-2} \ \mathrm{kg \ m^{-2}}$, almost all the extinction is generated by particles with $D_{\mathrm{g}}$ between $3 \times 10^{-4}$ m and $5 \times 10^{-3}$ m, in other words particles with a maximum dimension of around 1 mm. This corresponds

both to the peak in the mass-weighted PSD and the peak in the per-mass extinction. This peak in per-mass extinction could be called the "resonance" zone: particles with sizes that are a little larger than the wavelength give particularly large extinction even without normalisation by mass (see e.g. Petty, 2006, their Fig. 12.4). The ARTS plate aggregate is a less dense particle and the resonance zone is found at larger particle sizes. Hence the size range contributing to the bulk extinction is between $3 \times 10^{-4}$ m and $1 \times 10^{-2}$ m. The extension of the PSD to slightly larger particles (because the plate aggregate model implies

different parameters in the mass-size relation) also contributes to this. But the range of particle sizes which contribute to the bulk extinction is much smaller than the range of the mass-weighted PSD. In other words, there is a significant amount of particle mass with sizes smaller than 1 mm that is mostly or completely "invisible".

Within the Rayleigh scattering regime it is broadly the mass of ice, and not the particle shape, that controls the single-particle scattering properties[2]. For example, Fig. 12 of Eriksson et al. (2018) shows optical properties of non-spherical ice particles

converging for $x_{\mathrm{e}} < 0.5$, where the size parameter $x_{\mathrm{e}} = \pi D_{\mathrm{e}}/\lambda$ is based on the effective (mass-equivalent) particle size, not the maximum dimension. This is a more relaxed definition of the Rayleigh regime than often suggested ($x_{\mathrm{e}} < 0.1$ is typical) but using this, the ARTS ICON hail particle departs the Rayleigh regime above $D_{\mathrm{g}} = 2.6 \times 10^{-4}$ m at 190 GHz. However, even in the small particle limit, the extinction per mass shown on Fig. 6a is different between the ICON hail and the plate aggregate. This is a potentially confusing aspect of using the maximum dimension $D_{\mathrm{g}}$ as the x-coordinate. Because the ICON

hail particle is significantly denser, it thus has higher mass for the same $D_{\mathrm{g}}$ and hence even after normalisation by particle mass, it still has a disproportionate effect on the radiation field for the same particle size $D_{\mathrm{g}}$. Hence even in the Rayleigh regime, particle morphology still needs to be taken into account when mapping from particle size $D_{\mathrm{g}}$ to particle mass; this is not a completely obvious point given that Rayleigh and Mie sphere optical properties are typically described in terms of sphere diameter, rather than mass. This also further illustrates the importance of the mass-size relation of the particle model (Eq. 5;

Table 3) in determining the bulk optical properties. Interestingly, in the mass-weighted viewpoint of Fig. 6b, changing from the

---

[2]A shape dependence is theoretically possible even for $x \ll 1$ and this is provided by Rayleigh-Gans theory - see for example Hogan et al. (2017). However, shape dependence does not appear in the small particle limit of DDA simulations of totally randomly oriented particles.

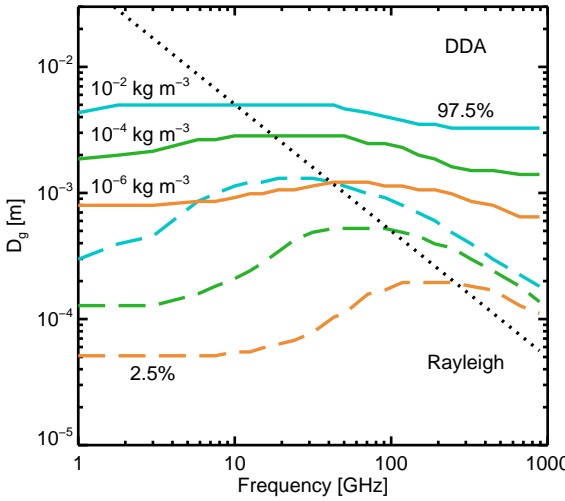

**Figure 7.** 2.5 (dashed) and 97.5 (solid) percentiles of contributions to the integration of bulk extinction (the "extagrand" in Figs. 6c and d) for the ARTS ICON hail particle and the F07 T PSD. These are shown as a function of frequency; other aspects of the integration are as Fig. 6. Water contents are coloured as indicated in the legend. The dashed black line indicates the limit of the Rayleigh regime for this particle; above this, non-spherical optical properties must be computed using DDA or equivalent method.

mass-size relation of hail (exponent $b = 2.99$) to that of the plate aggregate ($b = 2.26$) has only a secondary effect on the PSD shape.

A minor issue is that some particles with an exponent in the mass-size relation $b < 3$ (Eq. 5; Table 3; Fig 3) can be "super-dense" at small sizes, in other words that the mass-size relation implies a particle density that is higher than that of solid ice. This affects the plate aggregate (but not the ICON hail) below around $D_g = 2.7 \times 10^{-5}$ m. However, in the computation of the single-particle optical properties, whether Mie theory or DDA, particle densities are in practice not allowed to exceed those of solid ice. This could in theory result in an incorrect calculation of bulk extinction, but Figure 6 shows that even for the smallest water contents, any issue with representing superdense particles is irrelevant from the point of view of the optical properties, since particles as small as these are invisible. However, the treatment of small particles does affect the distribution of mass within the PSD, and hence can affect the bulk optical properties through renormalisation as explored in Sec. 3.2.2.

Figure 7 explores the frequency dependence of the range of particle sizes that are optically relevant; this is based on the ARTS ICON hail in order to show a particle which departs the Rayleigh regime at relatively low frequencies, here 10 GHz. The transition to non-Rayleigh scattering is associated with nearly an order of magnitude increase in the minimum particle size that contributes to the bulk optical properties (indicated by the 2.5 percentile of the integration here). However (and as also suggested by Fig. 6) the maximum particle size is more controlled by the PSD, and hence also the water content, and is less variable with frequency. This means that the size range contributing to the bulk optical properties is particularly squashed in the "resonant" region, which occurs just above the Rayleigh regime. One broad conclusion is that sophisticated models for





non-spherical particle scattering are always required to correctly simulate ice hydrometeor optical properties at microwave and sub-mm frequencies, even for very small water contents, and even for PSDs that do not generate such large particles as the

F07 T PSD (compare Fig. 4). The regions of the frequency and particle size spectrum where an approximate solution (such as Rayleigh or Mie) would be valid are those where the particles would be mostly invisible anyway. The ICON hail is the most dense available ARTS particle (Fig. 3) and would generate the most scattering from particles that are small in $D_g$. Hence this confirms that sub-100 $\mu$m ($D_g < 1 \times 10^{-4}$ m) ice particles are irrelevant to the radiative transfer up to 886 GHz. For an alternative view, excluding the constraint from the PSD, but finding similar results, see Pfreundschuh et al. (2020, their Fig. 5).

### 3.3   Representing preferentially oriented particles

The Liu and ARTS particles available in the hydrotable generator (and obviously the Mie sphere) represent only randomly oriented particles. However, ice hydrometeors are often preferentially oriented, as revealed by polarisation signatures in the high-frequency channels of microwave imagers (e.g. Defer et al., 2014; Gong and Wu, 2017). The ARTS database has recently been extended with a more advanced representation of particle orientation, giving particles a preferred canting angle, but

retaining random orientation in azimuth (Brath et al., 2020). However, this generates optical properties that are fully polarised, i.e. scattering can transfer energy from one polarisation to another. To model such fully-polarised optical properties would require the whole Stokes vector to be represented in the radiative transfer, but this is not available in a scalar fast model like RTTOV-SCATT. However, it is still possible to represent much of the effect of preferential orientation on microwave imager brightness temperatures, using an approximate method. This is done by scaling the bulk extinction $\beta_e$, as generated from fully

randomly oriented particles, according to a polarisation ratio $\rho$ (Barlakas et al., 2020):

$$\rho = \frac{\beta_{e,H}}{\beta_{e,V}} = \frac{(1+\alpha)\beta_e}{(1-\alpha)\beta_e} = \frac{(1+\alpha)}{(1-\alpha)} \tag{19}$$

Here, $\beta_e$ is increased by the proportion $\alpha$ to provide the scattering coefficient for use in horizontally (H-) polarised channels $\beta_{e,H}$. Similarly, it is reduced by $\alpha$ to provide the scattering coefficient for vertically (V-) polarised channels $\beta_{e,V}$. This description is more than just a tuning factor: $\alpha\beta_e$ describes the bulk scattering cross-section for linear polarisation in fully-polarised

radiative transfer, which represents the differences in the extinction between V- and H-channels (Barlakas et al., 2020).

Barlakas et al. (2020) found a polarisation ratio of $\rho = 1.4$ gave good reproduction of observed polarisation signatures at 166 GHz. Hence this approach is now implemented by default inside the main RTTOV-SCATT code. This operates on-the-fly and modifies optical properties stored in lookup tables using the standard "condensed" unpolarised representation, i.e. where optical properties are specified once per frequency, not per channel. In this approach a single polarisation ratio is applied

to all frozen hydrometeors. However, RTTOV-SCATT can also accept lookup tables that are specified once per channel (the "full" representation), and the table generator provides an option to generate polarised optical properties. The polarisation scaling from Eq. 19 is applied to H- and V- channels and is specified by an $\alpha$ which is a function of hydrometeor type, giving additional flexibility over the mechanism built into RTTOV-SCATT. However it is not yet possible to specify $\alpha$ as a function of frequency. The polarisation ratio approach works best for conical microwave imagers with zenith angles around 50° (Barlakas

et al., 2020). Cross-track sounders, which have polarisation and zenith angles that vary with scan position, are not yet supported





and will require further scientific development. Further, an approach for radar backscattering needs to be developed. The "full" channel representation provides a framework for future support of single-particle optical property databases based on oriented particles (e.g. Brath et al., 2020).

## 4   Effect of optical properties on brightness temperatures

### 4.1   Standardised two-stream cloud model

Although the bulk optical properties (e.g. Fig. 2) are already informative, their effect on radiation fields is both situation-dependent and a complex function of the optical properties. For example the effect of scattering on cloud-top brightness temperatures depends not just on the scattering coefficient but also the phase function, as summarised here by the asymmetry parameter. Further, even a relatively small amount of thermal emission within a cloud can substantially increase its brightness

temperature compared to a purely scattering case. Hence there is a need for a standardised and simplified way to compare the cloud-top brightness temperatures arising from different choices in computing the bulk optical properties. To do this we use the two-stream solution for the radiance at the top of a uniform cloud layer taking into account both scattering and thermal emission/absorption within the cloud (Appendix B):

$$I^{\uparrow}(\tau = 0) \quad = \quad I_0 \left[ \frac{1 - r_\infty^2}{\Phi} \right] + B_0 \left[ 1 - \frac{1 + r_\infty \left( \exp(\Upsilon \tau^*) - \exp(-\Upsilon \tau^*) \right) - r_\infty^2}{\Phi} \right]; \tag{20}$$

$$\Phi \quad = \quad \exp(\Upsilon \tau^*) - r_\infty^2 \exp(-\Upsilon \tau^*); \tag{21}$$

$$\Upsilon \quad = \quad 2\sqrt{1 - \omega_0 g}\sqrt{1 - \omega_0}; \tag{22}$$

$$r_\infty \quad = \quad \frac{\sqrt{1 - \omega_0 g} - \sqrt{1 - \omega_0}}{\sqrt{1 - \omega_0 g} + \sqrt{1 - \omega_0}}; \tag{23}$$

$$\tau^* \quad = \quad \beta \Delta z. \tag{24}$$

Here, $I^{\uparrow}(\tau = 0)$ is the upwelling radiance at the top of the cloud layer, which is assumed isotropic within each hemisphere

in the two-stream approximation. The vertical coordinate is optical depth $\tau$ which is 0 at the top of the cloud and $\tau^*$ at the bottom, so $\tau^*$ is the optical thickness of the cloud. $\Delta z$ is the geometric thickness of the cloud. The downwelling radiation at the top of the cloud is 0, and there is a steady source of upwelling radiation at the bottom of the cloud, $I^{\uparrow}(\tau^*) = I_0$. The upwelling radiance at the top of the cloud is given by Eq. 20 and is made up of below-cloud radiation that has been scattered or directly transmitted (the $I_0$ term) plus thermal emission from within the cloud, either scattered or directly transmitted to the

top of the cloud (the $B_0$ term, where $B_0$ is the Planck function at the temperature of the cloud, which is uniform throughout). The additional terms $\Phi$, $\Upsilon$ and $r_\infty$ are dependent only on the cloud's geometric thickness and the basic optical properties: the extinction $\beta_{\mathrm{e}}$, the SSA $\omega_0$ and the asymmetry parameter $g$ from the lookup tables. The terms $\Phi$ and $\Upsilon$ do not have a particular geophysical interpretation, but $r_\infty$ is the cloud-top albedo, of most relevance to solar radiation. The emitted radiation at the top of the cloud is hence a function of the three optical properties, plus the below-cloud upwelling radiation $I_0$, thermal emission

inside the cloud $B_0$ (and thus the cloud temperature) and the geometric thickness of the cloud $\Delta z$.

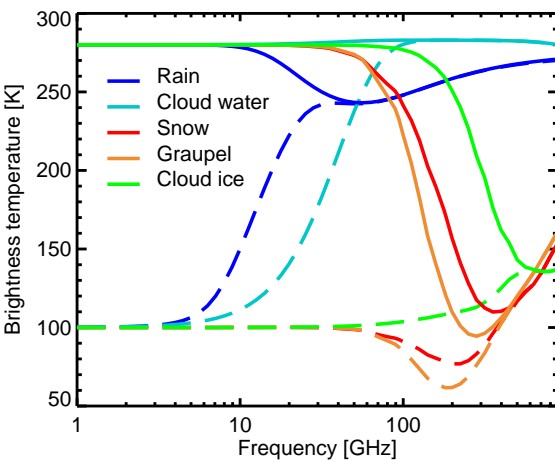

**Figure 8.** Cloud-top brightness temperatures simulated from uniform slabs composed of one of the default hydrometeor types (see legend, also Table 1) present in a 2 km thick layer with a water content of $l = 1 \times 10^{-3}$ kg m$^{-3}$. The cloud temperature is 253 K if frozen (snow, graupel or cloud ice) or 283 K if melted (rain, cloud water). Upwelling brightness temperature below the cloud is 280 K (solid lines) or 100 K (dashed lines).

The cloud just described is an approximate but compact description of typical situations in microwave and sub-mm radiative transfer. Gas absorption and emission have been neglected and the bottom boundary of the cloud is assumed black, so that any radiation leaving the cloud downwards can be forgotten – for example radiation reflected from the surface is ignored. This would still be a good representation of a cloud in the upper troposphere in any part of the spectrum where water vapour or

oxygen absorption blocks visibility of the surface, yet is not a significant source of emission at the level of the cloud itself. It is also a good representation of radiative transfer over land surfaces, where the surface is mostly black. It would be trivial to add gas absorption within the cloud and the surface-reflected term could be included but with additional complexity. But these would be a distraction from the simple standardised comparison of hydrometeor optical properties that is intended.

### 4.2   Overview of hydrometeor choices

Figure 8 shows the cloud-top brightness temperatures for standard two-stream clouds composed of one of the default hydrometeors from Table 1. In each case the cloud is 2 km thick and the water content $l = 1e^{-3}$ kg m$^{-3}$. This is quite a heavy cloud of 2 kg m$^{-2}$, but this is helpful in more clearly differentiating the types of hydrometeor. The cloud temperature is 253 K if frozen (snow, graupel or cloud ice) or 283 K if melted (rain, cloud water). A below-cloud upwelling brightness temperature of 280 K could represent a window channel over land or a lower-peaking water vapour channel (solid lines). In this case, scattering from

rain generates brightness temperature depressions peaking at around 40 K, and starting above 10 GHz. Cloud water is strongly absorbing, but the situation has minimal thermal contrast, so it provides only a tiny boost to brightness temperatures. Scattering





from the frozen hydrometeors is much more effective, generating depressions up to 200 K above 50 GHz for snow and graupel, and above 100 GHz for cloud ice.

In Fig. 8, a below-cloud upwelling brightness temperature of 100 K (dashed lines) could represent a window channel over
ocean (at lower frequencies, in horizontally polarised channel, ignoring the surface reflection) or a cirrus cloud above a very strongly scattering cloud placed lower in the atmosphere. Here, thermal emission from the rain and cloud water is the main effect above around 5 GHz. Above around 50 GHz, the frozen hydrometeors become visible, with snow and graupel generating up to 30 K brightness temperature depression even below the 100 K of upwelling radiation. The standard "graupel" configuration is a little more scattering than the "snow" particle, as intended (Geer, 2021b). The effect of cloud ice in this scenario
is to warm the brightness temperatures, since cloud ice has relatively low SSA. This warming effect of, for example, cirrus over a strongly scattering lower cloud has surprised a number of investigators (e.g. Xie et al., 2020; Barlakas et al., 2020). Where the dashed and solid lines join is where the clouds become optically thick and where below-cloud radiation becomes irrelevant. An interesting feature is that around 300 GHz to 500 GHz, snow and graupel clouds produce their lowest scattering TB depressions. Above this frequency of maximum scattering, these clouds start emitting more radiation again and brightness
temperatures are higher.

Figure 9 shows brightness temperatures for the standardised ice or snow cloud using the F07 tropical PSD with all options from the ARTS and Liu databases. The amount of brightness temperature depression from the ARTS particles roughly follows the amount of extinction in Fig. 2 and hence the progression broadly from low density aggregates and snowflakes to dense rimed particles. The Evans snow aggregate is again the least scattering, with TB dropping only to 150 K, and again the ICON
hail is the most scattering particle, dropping to 90 K. The frequency of maximum scattering varies from around 500 GHz for the Evans snow aggregate to around 200 GHz for the ICON hail. Between the extremes of light aggregate and dense hail, ARTS database provides a good spectrum of potential scattering properties. As mentioned before, the Liu database has bigger gaps and some of the particle shapes are almost redundant. The least-scattering Liu particle is the dendrite snowflake, but this still generates a relatively deep TB depression (as low as 130 K at 340 GHz). By contrast the ARTS database provides the column
aggregate and Evans aggregate, which produce even less scattering. This provides an important new capability to reproduce cloud ice signatures. Hence the ARTS column aggregate was required in the study of Geer (2021b) and is used in the new default RTTOV-SCATT configuration (Table 1).

At the strong scattering end in Figure 9b, the Liu columns and thick hex plate are more strongly scattering than the ICON hail from the ARTS database, giving TBs as low as 70 K around 180 GHz. Hence these Liu shapes continue to provide a capability
that is not available from the ARTS database. However it is interesting that the biggest variations in brightness temperature depression are in frequencies below 200 GHz. The studies of Geer and Baordo (2014) and Geer (2021b), based on SSMIS observations, have already benefitted from a spectral region of great sensitivity to particle habit and PSD. In these studies, the most strongly scattering particle models, those which generate deep TB depressions even at 50 GHz, have been decisively rejected, at least as a means of representing the effect of snow on TBs representing a global model grid box on the 10 km –
100 km scale.



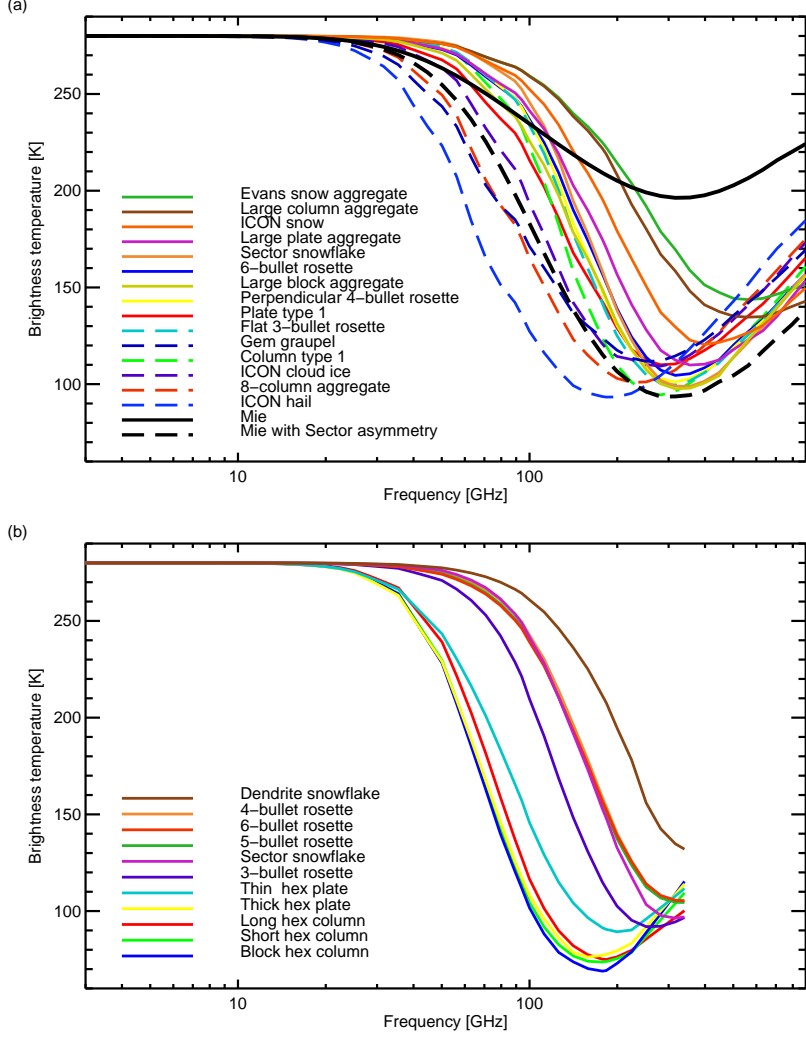

**Figure 9.** Cloud top upwelling brightness temperatures representing a standard ice or snow cloud at 253 K with upwelling below-cloud TB of 280 K: (a) ARTS database shapes, and a Mie snow sphere with details as Figs. 1 and 2: (b) Liu (2008) particles. In all cases the F07 tropical PSD has been used and the cloud layer is 2 km thick with a water content of $l = 1 \times 10^{-3}$ kg m$^{-3}$, as Fig. 8. The line and colour scheme for ARTS shapes is consistent with Fig. 2. Legends in this figure are ordered by 183 GHz TB, from highest to lowest.

Now is a good point to tidy up some loose ends from Geer and Baordo (2014), who rejected the Mie soft sphere as a viable model for snow particles at microwave frequencies. This was partly based on the excessive scattering below 100 GHz, in common with the dense non-spherical particles just described. However the soft sphere also provided insufficient TB depressions at 183 GHz; it is very clear in Fig. 9a, that it is an outlier compared to more-realistic non-spherical particle models. Geer and Baordo suggested that the primary problem of the soft sphere was not its overall level of extinction or scattering, but its unusually high asymmetry parameter, and hence stronger forward scattering (see Fig. 2: at 183 GHz and $l = 1 \times 10^{-3}$ kg m$^{-3}$, the





asymmetry of the soft sphere is around 0.9, compared to around 0.3 for the ARTS sector snowflake). The hypothesis was not confirmed in the study, but the standardised cloud model helps to clarify. The black dashed line in Fig. 9a shows TBs generated using the Mie bulk optical properties but with the asymmetry parameter from the ARTS sector snowflake. This hybrid parti-
cle gives a TB signature that is similar in some parts to the ARTS 8-column aggregate, making it indistinguishable from the DDA-based non-spherical particles. Hence it is the strong forward asymmetry of the Mie soft sphere that makes the difference. This result shows the utility of the standardised cloud model for assessing bulk optical properties, and the importance of the asymmetry parameter (and more broadly the phase function) in determining the brightness temperature.

## 4.3 Optical behaviour of frozen hydrometeors in the microwave and sub-mm

Figure 10 explores the sensitivity of the brightness temperatures to the cloud thickness, showing clouds of $0.2\,\mathrm{km}$ and $10\,\mathrm{km}$ geometric depth, respectively $0.2\,\mathrm{kg\,m^{-2}}$ and $10.0\,\mathrm{kg\,m^{-2}}$ integrated water content, and roughly representing frontal ice cloud compared to tropical deep convection. These can also be compared to Fig 9a with the "standard" cloud of $2\,\mathrm{kg\,m^{-2}}$. For the thin clouds, the deepest scattering TB depressions are at high frequencies, for example at $600\,\mathrm{GHz}$ for the large block aggregate. There is reasonable sensitivity to particle shape across all the higher frequencies, with a $50\,\mathrm{K}$ difference between the most and
least scattering shapes. For the thick clouds, TB depressions move to lower frequencies and get deeper still, with the ICON hail giving at TB of $50\,\mathrm{K}$ at $100\,\mathrm{GHz}$. At these mid-frequencies, the sensitivity to the particle shape of thick clouds is very strong, with around $150\,\mathrm{K}$ difference between the most and least-scattering particle. This frequency of maximum TB depression is not fixed, however, and increases to $350\,\mathrm{GHz}$ for the same particle in the thin cloud case. Conserving the integrated water content, but changing the water content by an order of magnitude in either direction (e.g. $l = 1 \times 10^{-2}\,\mathrm{kg\,m^{-3}}$ or $1 \times 10^{-4}\,\mathrm{kg\,m^{-3}}$, not
shown) the broad layout of these figures remains similar. Three interesting things appear above the frequency of maximum TB depression: First, the brightness temperatures start to become less sensitive to particle shape (and hence also particle size) which is particularly evident in the clustering on Fig. 10b; Second, there is an inversion in the ordering of scattering, with the dense particles (pristine crystals, hail and graupel) generally giving slightly warmer brightness temperatures than the less-dense aggregates and rosettes; Third, sensitivity to the integrated water content starts to disappear, with the thin cloud giving similar
brightness temperatures at $900\,\mathrm{GHz}$ to the thick cloud.

To further explain the transition in cloud optical properties as the frequency increases, we can return to Eq. 20. The cloud brightness temperature results from two terms: radiation transmitted by the cloud, and radiation emitted by the cloud. The terms in square brackets multiplying $I_0$ and $B_0$ are hence the transmittance and the emissivity of that cloud. These terms are shown in Fig. 11 along with the total brightness temperature. This is based on the strongly-scattering ARTS ICON hail particle with
the F07 T PSD, in order to show as much as possible of the transition within the available frequency range. The transmittance of the $2\,\mathrm{km}$ thick cloud goes to zero towards $1000\,\mathrm{GHz}$, in other words radiation from below can no longer pass through the cloud. However, the total brightness temperature bottoms out at around $100\,\mathrm{K}$, rather than dropping to zero, due to increasing thermal emission from within the cloud. As illustrated in Fig. 1, albeit with a different particle type, the single scattering albedo never reaches 1 (which would imply complete scattering or zero thermal emission); instead it reaches a maximum somewhere
around $500\,\mathrm{GHz}$ and starts to drop off at higher frequencies. This, along with the steady increase in overall extinction, must

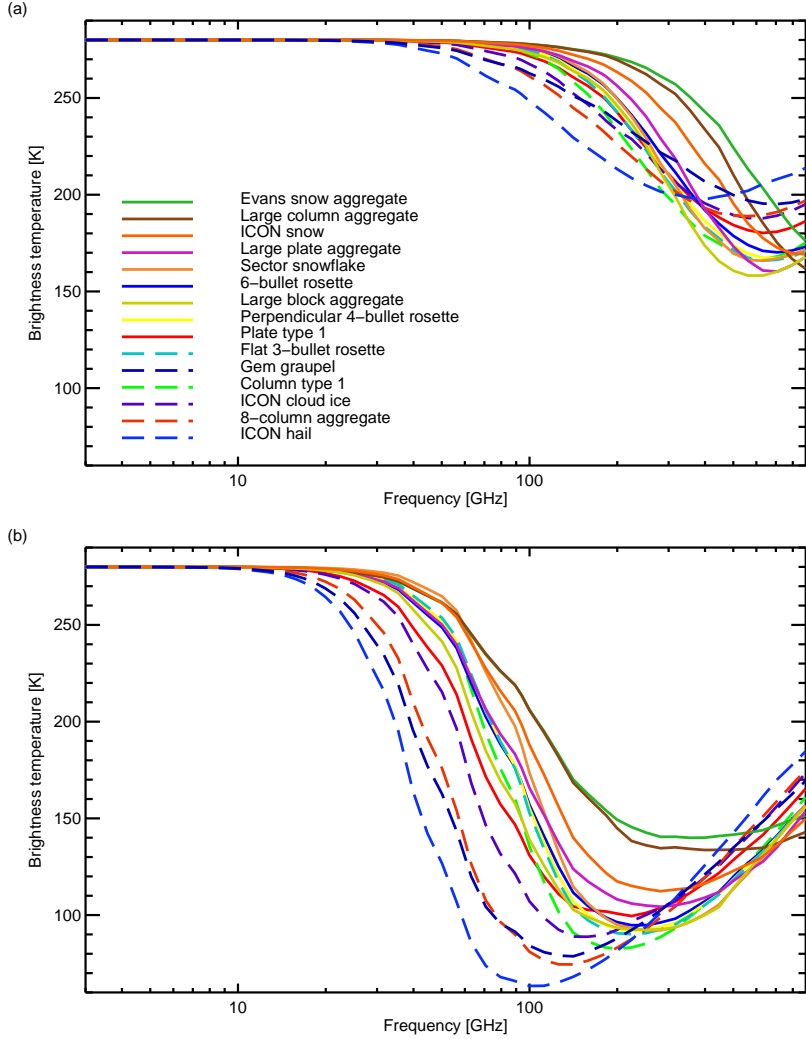

**Figure 10.** Brightness temperatures from a standard snow or ice cloud; details as Fig. 9a, including a fixed water content of $l = 1 \times 10^{-3}$ kg m$^{-3}$, but with a cloud geometric thickness of: (a) 0.2 km; (b) 10 km.

contribute to the continuous increase in thermal emission as the frequency increases. The clouds do not need to be optically thick to show this effect; Fig. 11 also shows the 0.2 km cloud, which even at 884 GHz has a transmittance of 0.5, but still has sufficient thermal emission to contribute significantly to the brightness temperature. Hence the spectral region around 500 GHz seems to be the turnaround point beyond which cloud optical behaviours tend towards those more familiar from the infrared, with increasing optical thickness, and cloud emission becoming the dominant optical process.

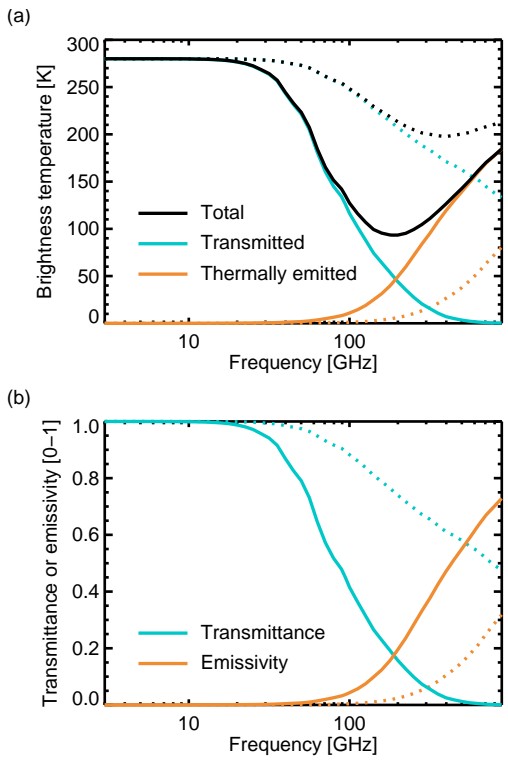

**Figure 11.** (a) Brightness temperatures, divided into the transmitted and emitted parts of Eq. 20; (b) Transmittance and emissivity of the same cloud, i.e. the terms in square brackets in Eq. 20. This is a standard cloud with a thickness of 2 km (solid) or 0.2 km (dotted), with the ARTS ICON hail particle, F07 T PSD, and other settings as seen in Figs. 9 and 10.

## 4.4 Sensitivity to PSD

Figure 12 explores the sensitivity to alternative PSDs, with the representation of ice cloud in mind. The Heymsfield et al. (2013) PSD (top panel) provides quite similar results to F07. Although the order of scattering varies slightly in the detail, compared to Fig. 9a, the main features of the ARTS particles are preserved. MH97 generates a more compact spread of
brightness temperatures across the ARTS particles, but it does not much change the overall order of scattering. The Evans snow aggregate, for example, is still the least-scattering of the particles. This compact spread is attributed to the MH97 PSD putting a relatively high proportion of particles in the smaller size ranges (Ekelund et al., 2020b) and hence keeping more of the particles in the Rayleigh regime. However the bunching of the different particle models still occurs above 500 GHz, suggesting this is not strongly affected by the particle size. Hence this supports a fairly universal transition towards an emission-dominated,
optically thick regime above 1000 GHz, where sensitivity to particle size and shape appears to become smaller.

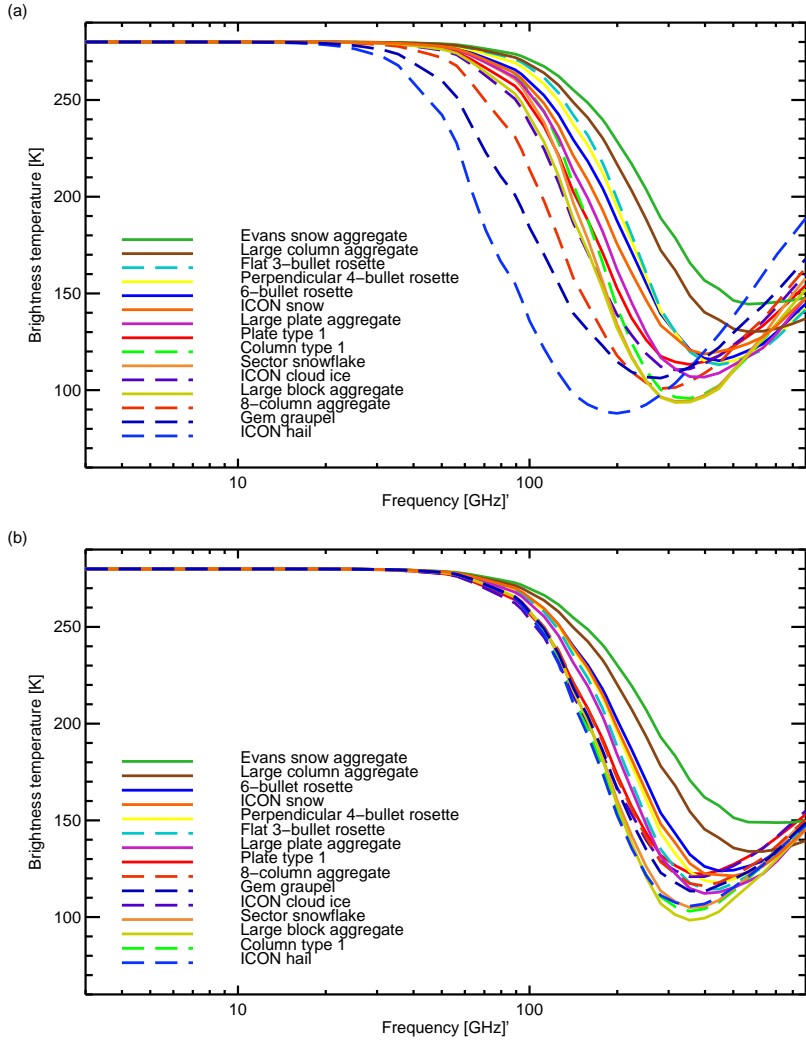

**Figure 12.** Brightness temperatures from a standard snow or ice cloud; details as Fig. 9a but exploring different PSDs: (a) Heymsfield et al. (2013, H13); (b) McFarquhar and Heymsfield (1997, MH97).

## 5 Conclusion

This work has summarised the process of generating bulk hydrometeor optical properties based on physical assumptions about the sizes, masses, habits and orientations of cloud and precipitation particles. It documents the hydrotable generator for microwave and sub-mm scattering radiative transfer in version 13.0 of RTTOV (Radiative transfer for TOVS, Saunders et al., 2018, 2020), a widely used satellite simulator, hopefully with relevance to the users of many similar tools. The work has overviewed the bulk optical properties and brightness temperatures generated by Mie spheres and two databases of non-spherical ice particles (Liu, 2008; Eriksson et al., 2018).




A focus has been the existing and newly supported particle size distributions (e.g. McFarquhar and Heymsfield, 1997; Petty and Huang, 2011; Heymsfield et al., 2013) and the core process of numerical integration across them. This process maps from a chosen particle size distribution and particle model, through to individual particle masses and other physical properties, and finally through to bulk optical properties. This process has a number of issues. For example, very small ($< 100 \mu$m) particles are invisible at these frequencies but can affect the bulk optical properties through renormalisation adjustments to the PSD, relating back to the underlying question whether current PSDs do a good job of representing small particles (Korolev et al., 2011; O'Shea et al., 2020). Furthermore, the effect of small differences in the particle mass-size relation has been highlighted in the results using the ARTS and Liu sector snowflakes, which have exactly the same single-particle optical properties, but generate significantly different brightness temperatures due to slightly different mass-size relations.

To illustrate the available options, a standardised homogeneous layer cloud was proposed. This is based on a simple two-stream analytical solution and converts the bulk optical properties into brightness temperatures, which are easier to interpret. In particular this resolves the trade-offs between the absolute level of scattering (the scattering coefficient) and the asymmetry parameter (summarising the shape of the scattering phase function) in the amount of brightness temperature depression that is generated. A further aspect of the standardised homogeneous layer cloud is that it illustrates the balance between radiation coming into the cloud from below, that may be scattered or absorbed, versus thermal emission from the hydrometeors themselves. This is particularly important for frozen particles above $200$ GHz where, as we have shown, the cloud optical properties start to move towards an emission-dominated, optically thick regime that is much less sensitive to particle size and shape, a regime more familiar from the behaviour of clouds in the infrared.

The standardised cloud helps further investigate the soft Mie sphere as a representation of ice particles. This has unusually strong forward scattering (high asymmetry) compared to most non-spherical particle models representing the same ice mass (e.g Eriksson et al., 2015, their Fig. 3). This unusually strong asymmetry is also found in the bulk scattering properties. The standardised cloud model shows that the spherical model cannot generate sufficient brightness temperature depressions at higher frequencies. It generates a "scattering spectrum" that is very different to any more physically reasonable model. Just by replacing the asymmetry of the soft Mie sphere with that from a non-spherical model, it becomes a much more plausible representation falling within the range of non-spherical models. This is further proof, if any is needed, of the problems with the Mie sphere representation of snow; Kuo et al. (2016) have made a very similar point. Only within the Rayleigh regime do spherical and non-spherical particles generate similar optical properties as a function of the particle mass. However, even at $10$ GHz a significant fraction of ice particles can be outside the Rayleigh regime (Fig. 7); hence there really is no alternative to taking account of the microphysical characteristics of realistic non-spherical ice particles when simulating observations in the microwave and sub-mm regions.

An underlying assumption in this work (and many others that follow the same approach) is that clouds can be represented using a single particle shape model to cover all instances of a highly heterogeneous class of particles like snow - a "one shape fits all" approach. An important aspect of this approach is that the particle shape model defines the mass-size relation; this affects the bulk scattering properties both by its influence on the shape of the PSD and simply by defining the mapping between maximum dimension $D_{\mathrm{g}}$ and the particle mass, which means that particles of nominally the same size can generate



very different single-particle optical properties. This applies even within the Rayleigh regime. The use of mass-equivalent diameter $D_e$ could remove the latter issue (e.g. Eriksson et al., 2015) but this is of less practical relevance as long as the PSDs

which are used to map from water content to particle size (and hence mass) are defined in terms of maximum dimension $D_g$. A more physically-based approach would be to consider an ensemble of particle habits (e.g. Kulie et al., 2010; Baran et al., 2011), and might impose a mass-size relation thought to be an appropriate description of certain hydrometeors, such as midlatitude stratiform ice cloud (e.g. Brown and Francis, 1995; Hogan et al., 2012). However, RTTOV has the job of modelling satellite observations with reasonable accuracy across the entire globe; microwave observations are strongly sensitive to many different

cloud regimes and particularly to those such as deep convection, where even basic details of the microphysics remain poorly known. Forecast models are currently incapable of representing the full range of microphysical parameters needed to constrain the hydrometeor optical properties; therefore simplification and generalisation is unavoidable, with parameter estimation being the ideal method for identifying the best microphysical assumptions (Geer, 2021b, with an appropriate Bayesian framework this could incorporate prior expert microphysical knowledge, such as the realism of particle habits, PSDs and mass-size relations).

In the event that a much richer microphysical representation is required, RTTOV-SCATT allows an unlimited number of hydrometeor categories that could be used to represent almost any level of microphysical complexity. Hence the tools are already available, but the underlying issue is whether is possible to appropriately specify this complex microphysics.

There are many ways to further improve the representation of bulk scattering from hydrometeors, most of them applicable generally:

– RTTOV (and many other codes) use a different mechanism and different physical assumptions for the optical properties of clouds in the IR and solar regions. A major future development should be to produce optical properties from the microwave to the UV using the same approaches as much as possible; this requires non-spherical scattering databases to be expanded to cover the whole range, a process that is only starting (e.g. Yang et al., 2013; Ding et al., 2017; Baran et al., 2018).

– More work is needed to unpick the tight coupling between the particle shape model, mass-size relation and the particle size distribution. Better ways of mapping bulk hydrometeor mass to particle ensembles would be very welcome.

– More work needs to be done to standardise and/or interface the physical assumptions, such as shape, PSD and mass-size relation, with assumptions made in atmospheric models. This would particularly support efforts to learn better physical models of cloud and precipitation, directly from observations (e.g. Schneider et al., 2017; Geer, 2021a).

– A global effort is ongoing to standardize and package many of the different scattering databases that are becoming available (see Kneifel et al., 2018); once complete, interfaces will be added to allow the user a much broader choice of particle models. This would allow the use of non-spherical oriented raindrops, for example (e.g. Ekelund et al., 2020a).

– Current PSDs have significant limitations, such as the small-particle bulge that is now thought to be unphysical (Korolev et al., 2011; O'Shea et al., 2020). Even modern PSDs based on large amounts of aircraft data, like Heymsfield et al.



(2013), provide large numbers of large (cm-sized) particles so in RTTOV-SCATT they do not provide a good representation of the "cloud ice" category in global models (Geer, 2021b). As with the simple exponential PSD now used as the default for cloud ice, it may in some cases be better to infer PSDs through parameter estimation, using the parameters of the standardised Modified Gamma Distribution (MGD, Petty and Huang, 2011).

– A representation of non-spherical melting hydrometeors (e.g. Johnson et al., 2016; Leinonen and von Lerber, 2018) needs
to be included. This will be particularly important for simulating radar backscatter in the melting layer (the bright band). It would also improve brightness temperature simulations, which may otherwise be too low in channels and situations sensitive to the melting layer, possibly by up to 8 K (Bauer, 2001).

This survey has also revealed a few issues more specific to the RTTOV hydrotable generator:

– The lowest temperature bin, at 203 K, is at least 20 degrees higher than the lowest tropospheric temperatures. Hence, in a
future version, it is necessary to extend the lower temperature range of frozen particles permitted by the table generator. Note that, although the lowest temperature point in the ARTS database is 190 K, Eriksson et al. (2018) showed that extrapolation would be reasonably accurate down to 150 K.

– The default representation of rain has not been updated since Bauer (2001); apart from the aforementioned points on shape and orientation, it is worth examining how well the rain PSD is represented, and potentially updating this.

– The code is operated offline, with a lookup table approach, but it is fast enough to be operated online within the radiative transfer code, especially as weather forecasting systems get more capable (e.g. English et al., 2020). This will be an aim of future development work.

To summarise, the future evolution of the code may be towards online operation, so that parameters of the PSD and even particle shape can be estimated directly from the observations, rather than prescribed in a "one shape fits all" approach.

*Code and data availability.* This work is based on RTTOV v13.0, which is available for free from https://nwp-saf.eumetsat.int/ to users who have registered and agreed to the licence conditions. The licence is available at https://nwp-saf.eumetsat.int/site/software/licence-agreement/ with the main condition being that onward redistribution of the code is not permitted. For the purposes of anonymous review of the current manuscript, the code of the hydrotable generator is temporarily available at http://nwp-saf.eumetsat.int/downloads/james/rttov13_hydrotable_gen_for_review_only.tar.gz. This code should not be redistributed or used for any other purpose.

*Author contributions.* All authors contributed to the code, science and data embedded in the RTTOV-SCATT hydrotable generator. AG wrote the majority of the text, with reviews and contributions from all other authors.

*Competing interests.* No competing interests are present



*Acknowledgements.* EUMETSAT NWP-SAF are acknowledged for funding and coordinating the overall RTTOV development. The module described in this article was developed with contributions from many different scientists and funding streams. Vasileios Barlakas is funded

by a EUMETSAT fellowship. Niels Bormann, Stephen English and Robin Hogan are thanked for reviews of the manuscript.

## Appendix A: Radar reflectivity

The radar reflectivity factor is computed as a scaling of the computed backscatter $\beta_{\mathrm{b}}$,

$$Z = \frac{10^{18}}{z_0} \beta_{\mathrm{b}}, \tag{A1}$$

where the factor $10^{18}$ converts from SI to the more typical units of [$\mathrm{mm}^6\ \mathrm{m}^{-3}$] and $z_0$ comes from the definition of the radar

reflectivity factor as the sixth moment of a notional distribution of liquid spheres that would produce the equivalent backscatter
(see Petty, 2006, for further details):

$$\beta_{\mathrm{b}} = \underbrace{\frac{\pi^6}{\lambda^4} \left| \frac{\epsilon_{\mathrm{water}} - 1}{\epsilon_{\mathrm{water}} + 2} \right|^2}_{z_0} \underbrace{\int\limits_0^\infty D_{\mathrm{g}}^6 n_{\mathrm{g}}(D_{\mathrm{g}})\, dD_{\mathrm{g}}}_{M_6 := Z} := z_0 Z. \tag{A2}$$

Here, by definition of the reflectivity factor, the backscatter cross-section is assumed to be given by the Rayleigh scattering
approximation ($\sigma_{\mathrm{b}}(D_{\mathrm{g}}) = z_0 D_{\mathrm{g}}^6$). $\epsilon_{\mathrm{water}}$ is the complex permittivity of liquid water at the radar frequency, at an assumed tem-

perature of 273 K, and $\lambda$ is the corresponding wavelength. Hence the reflectivity factor is dependent on the chosen liquid water
permittivity model (see Sec. 2.1). If these configuration choices are not sufficient, users would need to directly change the code
according to the way the radar reflectivity factor is defined for any particular instrument; this might need to be improved in
future.

## Appendix B: Two-stream slab cloud with scattering and thermal emission and absorption

The two-stream approximation for radiative transfer is proposed here as a way to compare different sets of bulk optical proper-
ties. To simplify the unpolarised full scattering radiative transfer equation, the two-stream approximation assumes the radiance
field is constant in each hemisphere (as a function of azimuth and zenith angle) and is hence described by just two variables, the
upward and downward radiances $I^\uparrow$ and $I^\downarrow$ (Thomas and Stamnes, 1999; Petty, 2006). A second assumption is that backscat-
tering from one hemisphere into another is proportional to the asymmetry parameter $g$. With these assumptions, the following

radiative transfer equations can be defined, which are averaged over all azimuth and zenith angles in each hemisphere:

$$\frac{1}{2} \frac{dI^\uparrow(\tau)}{d\tau} = (1 - \omega_0)(I^\uparrow(\tau) - B) + \frac{\omega_0(1-g)}{2}(I^\uparrow(\tau) - I^\downarrow(\tau)) \tag{B1}$$

$$-\frac{1}{2} \frac{dI^\downarrow(\tau)}{d\tau} = (1 - \omega_0)(I^\downarrow(\tau) - B) - \frac{\omega_0(1-g)}{2}(I^\uparrow(\tau) - I^\downarrow(\tau)) \tag{B2}$$

This follows the derivation in Petty (2006) but does not drop the Planck function $B$, so it represents thermal emission as well as
scattering. The single scattering albedo $\omega_0$, asymmetry $g$ and temperature (and hence Planck function $B$) are assumed constant





in optical depth $\tau$. Adding and subtracting Eqs. B1 and B2, then differentiating, and then substituting again from Eqs. B1 and B2, gives:

$$\frac{d^2}{d\tau^2}(I^\uparrow(\tau) + I^\downarrow(\tau)) = \Upsilon^2\left[(I^\uparrow(\tau) + I^\downarrow(\tau)) - 2B\right] \tag{B3}$$

$$\frac{d^2}{d\tau^2}(I^\uparrow(\tau) - I^\downarrow(\tau)) = \Upsilon^2(I^\uparrow(\tau) - I^\downarrow(\tau)) \tag{B4}$$

where $\Upsilon$ has been defined in Eq. 22. These equations have generally-known solutions, so:

$$I^\uparrow(\tau) + I^\downarrow(\tau) = c_4\exp(\Upsilon\tau) + c_2\exp(-\Upsilon\tau) + 2B; \tag{B5}$$

$$I^\uparrow(\tau) - I^\downarrow(\tau) = c_3\exp(\Upsilon\tau) + c_4\exp(-\Upsilon\tau). \tag{B6}$$

Here, $c_4$ to $c_4$ are coefficients that need to be found. Further manipulation, similar to Petty (2006), gives the solutions for the upward and downward radiances:

$$I^\uparrow(\tau) = A\exp(\Upsilon\tau) + r_\infty D\exp(-\Upsilon\tau) + B; \tag{B7}$$

$$I^\downarrow(\tau) = r_\infty A\exp(\Upsilon\tau) + D\exp(-\Upsilon\tau) + B. \tag{B8}$$

Here, $A$ and $D$ are constants deriving from $c_4$ to $c_4$ that still need to be determined. $r_\infty$ is the cloud albedo, a function of $\omega_0$ and asymmetry $g$ defined earlier. The constants are found by imposing the boundary conditions $I^\downarrow(\tau^*) = 0$ and $I^\uparrow(\tau^*) = I_0$, in other words zero downward radiation at the top of the cloud, and a fixed amount of radiation upwelling from below the cloud. The boundaries are hence assumed black bodies, unaffected by the radiation emitted from the cloud. This gives:

$$I_0 = A\exp(\Upsilon\tau^*) + r_\infty D\exp(-\Upsilon\tau^*) + B; \tag{B9}$$

$$0 = r_\infty A + D + B. \tag{B10}$$

These can be easily solved for $A$ and $D$, and the radiation emitted at the top of the cloud $I^\uparrow(0)$ can be determined from Eq. B7 to provide Eq. 20. Setting $B = 0$ recovers the two-stream pure-scattering example from Petty (2006, his equations 13.39 and 13.40) . Setting $I_0 = 0$ and $g = 0$ (no external radiation sources, isotropic scattering) recovers the "imbedded source" solution from Thomas and Stamnes (1999, their equations 7.69 and 7.70).





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
