# Peer review of "Bulk hydrometeor optical properties for microwave and sub-mm radiative transfer in RTTOV-SCATT v13.0"

_Geoscientific Model Development, 2021_

## Author Response (AR1)

Both reviewers are thanked for their helpful comments, copied below in italic. Author responses made during the discussion phase follow an arrow (->). Final comments on how the manuscript changed are in bold.

Reviewer 1

*eq 18 is weird:   âsigma(Dg)/m(Dg) xâm(Dg)  -- looks like the m(Dg) terms would cancel out?*

-> This is intentional, with the reasoning described in the text on lines 528 - 534: it is a normalisation used in Figure 6, and Eq. 18 is just Eq. 1 but multiplied and divided by the same factor, m(Dg). A possible improvement may be to add underbraces and captions for the two expressions that are created (the mass-normalised extinction cross-section sigma(Dg)/m(Dg), and the mass distribution m(Dg)n(Dg).

**Equation 19 is now modified using underbraces to better explain these two new terms.**

*Radar definition: Lines 855-860:  "Here, by definition of the reflectivity factor, the backscatter cross-section is assumed to be given by the Rayleigh scattering*
*approximation (âb(Dg) = z0D6g )"    This simplification shouldn't be necessary, except in that limiting case where it applies.    From the particle properties, you have all of the backscattering information necessary to compute the non-Rayleigh reflectivity.   Furthermore, for ice-phase hydrometeors, one would have to consider the modifications to the dielectric constant in the "z0" term, unless you're explicitly making the decision that all reflectivity factors are computed *as if* the reflecting hydrometeors are liquid water with an equivalent diameter.*

-> The latter is true, all reflectivity factors are computed as if the reflecting hydrometeors are liquid water. The "z0" term is a constant, based on Rayleigh scattering. The backscatter beta_b is still generated using the full capabilities of the hydrotable generator (e.g. non-Rayleigh scattering), it is then just scaled by z0/10^18 (equation A1). The form of equation A2 was intended as a compact description of the radar reflectivity factor but it seems it is confusing. A proposed remedy is to label the reflectivity defined in equation A2 as beta_b^Rayleigh, and then to better explain that the radar reflectivity is the Z for which beta_b (computed using the full scattering capabilities of the hydratable

generator) is equal to beta_b^Rayleigh, assuming Rayleigh scattering from a liquid sphere. This is how the observable is reported for the spaceborne radar instruments such as DPR and Cloudsat.

**Appendix A has been rewritten, and the equations have been tweaked, hopefully making it clearer for readers to understand how the radar reflectivity factor is calculated.**

*In figure 1, what are the effective diameters of the assumed size distributions for the various hydrometeor types given the fixed water content?   Similar comment for Fig. 2, particularly for consideration of the radar reflectivity, which is strongly dependent on the size of the particle.*

-> A consideration of the effective diameter (the area-weighted mean diameter, or equivalently the ratio of moments $M\_3/M\_2$) could add some useful information. Before creating the revised version of the paper, the effective radii will be calculated, and potentially included in the revised manuscript.

**It would have been very hard to fit a discussion of effective diameter into the existing text without an extensive rewrite, so a new appendix B and figure B1 has been added instead. This is referenced from the main text in section 3.2.4, which is where the main discussions about mass and geometric diameter are located. The appendix shows the effective diameter as a function of water content for all the ARTS particles and all the standard hydrometeor types, more as a reference than anything else, since the effective diameter is not a great predictor of microwave optical properties (this is already addressed from the point of view of single-particle optical properties in section 3.2.4).**

*Lines 274-275: "In Fig. 1 the frozen particles have small oscillations with frequency, particularly obvious in the radar reflectivity at lower frequencies. This is a result of interpolating away from the original temperature, size and frequency steps in the ARTS database."     This seems more like an error in the interpolation routine.  In every other study I've seen that provides similar plots, this behavior is not observed on interpolated points.*

-> The interpolation is carried out on the single-particle scattering properties in 3 dimensions, using the trilinear approach. The trilinear interpolation routine has been carefully checked. A first point is that bilinear and trilinear interpolation only give linear results from translation along one of the interpolation axes. Along a transect that is not parallel to one of the axes, a nonlinear function is generated. A second thing to point out is that what is shown are the integrated bulk hydrometeor optical properties, so they are the integrated results of many trilinear interpolations. In figure 1, for example, the temperature is held constant, the frequency is obviously changing (it is the x-axis of the plot) but also the third coordinate of the underlying interpolation (particle size) is also implicitly changing because at different frequencies the bulk optical properties are sensitive to different parts of the PSD. Hence the figure effectively shows a transect along a direction in which both particle size and frequency are changing, where it is not guaranteed to see linear behaviour from the interpolation. This is not the case in figure 2, for example, where both the frequency and the temperature are held constant in the underlying trilinear interpolations, and any variation is along the particle size axis, so this is guaranteed to give linear variations between tie-points in the single-particle optical property database. Admittedly, the manuscript could do better to explain what is actually quite an interesting effect; maybe even a new appendix could be added to give a more complete explanation.

**This discussion is now made in a short footnote**

*Equation 6: Is Dg equivalent to Dmax for non-spherical particles, and if how is this handled in the size-integration over various Dmax values, while maintaining the appropriate mass-dimension relationship?*

-> Here, Dg is defined as in Petty and Huang (2011) as the geometric size of the particle, and for non-spherical particles this is the longest dimension of the particle (line 86 of the current manuscript). This is also referred to in some papers as Dmax, so from that point of view Dg is equivalent to Dmax as used by other authors. However, in the current work Dmin and Dmax are used as the bounds of the integration across the PSD, so Dmax has a completely different meaning.

**No changes made**

*With regard to this as well, it wasn't clear to me (perhaps I missed it), but an accounting of the difference in density between bulk ice and liquid water need be made if one is using liquid-equivalent diameter as the baseline for all mass comparisons across hydrometeors. See, for example: Bennartz, R. and Petty, G.W., 2001. The sensitivity of microwave remote sensing observations of precipitation to ice particle size distributions. Journal of Applied Meteorology, 40(3), pp.345-364.*

-> This is true, but the equivalent diameter is only relevant in a few places in this work (e.g the description of the McFarquhar and Heymsfield (1997) PSD). In plots such as Figure 3 the focus is on frozen hydrometeors and comparisons are not being made between liquid and frozen hydrometeors.

**No changes made**

*Line 302, units for density are incorrect / inconsistent.*

-> thanks for spotting this, it was a mistake and the intended unit was kg m^-3

**Typo fixed**

*Eq 8 is the := intentional? Also, may want to mention that this is the "k-th moment" equation, and in later equations, "b = k".*

-> These are helpful suggestions for making the manuscript clearer; they will be followed in the revised manuscript. However, the := was intentional to indicate a definition of the moment M_k, but on reflection any use of the := will be removed from the manuscript as it might not be 100% clear, and if the equation is a definition this will be clearly stated in the text.

**All := were removed from the manuscript, the definition of PSD moments was split into a separate equation, and the relevant text was slightly rewritten.**

*Table 4: SI units?*

-> Yes, this will be added to the table caption

**Added to table caption**

*Section 3.3.2: You can use the incomplete gamma function solution to definite integrals, this is discussed in Petty and Huang 2011. Not a recommended change, just an observation.*

-> Interesting point, this could be useful for the future to avoid renormalisation. A sentence will be added in the revised manuscript.

**Sentence added.**

*Line 538, water content is typically expressed in k m^{-3} ? See e.g., Line 651. Perhaps "water path" would be a more appropriate term when referring to the vertically integrated content (even if for a single layer).*

-> Sorry, another typo. The intended unit was kg m^-3, and this was a water path.

**Typo fixed**

*Lines 550 area: There's some hand-wringing about Dg as the diameter variable of choice, but I think much effort would be saved using De instead. The logic is that De is a proxy for mass, by virtue of being the mass equivalent radius. This ensures equivalent mass comparisons in the PSDs, and greatly simplifies phase transitions such as melting / freezing. This is, however, not a recommendation for this paper, as it would require a complete overhaul of everything done so far.*

-> Agreed. Moving to a De basis for the integrations was an aspiration for the v13.0 developments, but there wasn't time to try it out. This is something that could be very useful in the future. However, RTTOV is moving towards a full-spectrum approach in the next few years, it is hoped in future to generate optical properties from the microwave to the visible from the same package. Hence it is worth consideration whether a De basis would also suit the higher frequencies where the particle's geometric cross section is more important.

**An extra bullet was added in the conclusions specific to the hydrotable generator.**

*Line 665: "this warming effect" is simply due to the fact that smaller particles are more emissive than scattering at higher frequencies.*

-> The text will be improved in this area to reflect this point better, i.e. that the primary difference between "cloud ice" and "snow" is the chosen PSD, rather than the particle shape, and so indeed it is the fact that particles are in general smaller in the cloud ice category.

**Additional sentence added at the end of the first paragraph of section 4.2 and also this point is clarified in section 2.2.**

*Line 669: "Above this frequency of maximum scattering, these clouds start emitting more radiation again and brightness 670 temperatures are higher."   True, but it's also relevant to point out that surface-induced polarization effects (e.g., over ocean) ALSO decrease with increasing optical depth -- this provides a separate key piece of information content.*

-> Although the use of dual-polarised channels is an important point for the wider information content of microwave radiances, it's not clear whether this refers to the (on average) increasing optical depth from water vapour as frequency increases through the sub-mm, or the point refers to the information coming from dual-polarised channels at a particular frequency. Since the discussion is based on a hypothetical slab cloud with non-poliarsed optical properties and a fixed warm or cold upwelling non-polarised brightness temperature from below, it is not intended to fully cover all aspects of the microwave and sub-mm information content. Perhaps the best way to address this comment is just to note in the definition of the slab cloud that it does not consider polarisation in any way.

**Additional sentence added in section 4.1**

*Reviewer 2*

*… the choice of colors used in the plots: Perhaps red and green should not be used together at the same time in consideration of the colorblind.*

-> This is a good point. The figures 2,3,9,10 and 12 do include both red and green lines, but there are so many lines they are hard to tell apart even with typical colour vision. However, the line keys have been carefully devised to follow the order of the lines at a certain point on the

figure. Hence it should be possible to identify a line by its relative position on the plot, even without using colour or line style. Figures 1, 4 and 8 may be more problematic, particularly the use of green for the cloud ice category and red for snow. Hence, these will be revised with an alternative colour (or possibly line style).

**Figures 1, 4 and 8 have been revised.**

*Additional changes in the revised manuscript*

**The launch dates of ICI have been changed, the recent Barlakas et al, Geer and O'Shea citations have been updated, and a few other typos corrected (e.g. kg m-2 vs. kg m-3), caption to Fig. 4, text in Fig. 5 (top panel is l=10-6, not 10-2).**

**An incorrect description of the way totally randomly oriented particles are represented in the Liu database has been corrected (section 2.1.3).**

**The statement that "mass is the most important predictor of microwave optical properties" is clearest if we already know what composition of particle we are discussing (ice or water) so this has been made clearer in the introduction to section 2.2.1 (note the discussion of whether composition or mass is "more important" is probably a bit difficult, and mostly irrelevant to the current manuscript).**

**The impact of SSA, extinction and absorption in generating the impact of ice cloud in Figure 8 has been explained a little more precisely (second paragraph of section 4.2)**

**A citation to Baran et al. (2014) has been added to the summary of efforts to produce consistent optical properties across frequencies (in the conclusion).**

**The link to the temporary code repository has been removed from the "Code and Data Availability" statement as it was only intended for review purposes. As discussed with executive editor Dr. Añel, the RTTOV development team has asked EUMETSAT to allow code archiving and DOIs. The latest in this process is that EUMETSAT is willing to start generating DOIs for RTTOV code versions, but it**

continues to investigate the options: either directly itself or through Zenodo. Unfortunately this process is ongoing and it is not yet possible to give a permanent software link in this section; just the reference to the RTTOV website.

A number of other small changes have been made, intending to improve clarity and readability throughout the manuscript, without any change to the intended meaning.